

# The equilibrium landscape of the Heisenberg spin chain

**Enej Ilievski[1]\* and Eoin Quinn[1,2]†**

**1** Institute for Theoretical Physics Amsterdam and Delta Institute for Theoretical Physics,
University of Amsterdam, Science Park 904, 1098 XH Amsterdam, The Netherlands
**2** LPTMS, CNRS, Univ. Paris-Sud, Université Paris-Saclay, 91405 Orsay cedex, France

\* e.ilievski@uva.nl, † eoin.quinn@u-psud.fr

## Abstract

We characterise the equilibrium landscape, the entire manifold of local equilibrium states, of an interacting integrable quantum model. Focusing on the isotropic Heisenberg spin chain, we describe in full generality two complementary frameworks for addressing equilibrium ensembles: the functional integral Thermodynamic Bethe Ansatz approach, and the lattice regularisation transfer matrix approach. We demonstrate the equivalence between the two, and in doing so clarify several subtle features of generic equilibrium states. In particular we explain the breakdown of the canonical $\mathcal{Y}$-system, which reflects a hidden structure in the parametrisation of equilibrium ensembles.

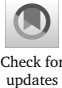

# 1 Introduction

The equilibration phenomena of quantum many-body systems have become a vigorous research topic for both theoretical and experimental studies of condensed matter systems in recent years. For generic interacting systems a central role has been played by the Eigenstate Thermalisation Hypothesis [1–3], which offers a unifying framework for characterising ergodic behaviour. The unconventional equilibration exhibited by (nearly) integrable systems has also drawn substantial interest, leading to the notion of the Generalized Gibbs Ensemble (GGE) [4–6]. The anomalous behaviour of integrable systems is due to the existence of infinitely many local conservation laws, whose essence is to protect quasi-particle excitations against decay [7]. The majority of the literature on equilibration in integrable systems has focused on non-interacting models, where the concept of a generalised Gibbs ensemble is synonymous to prescribing the occupations of single-particle modes [8,9]. *Interacting* integrable systems on the other hand exhibit rich spectra of stable excitations which undergo non-trivial completely factorizable scattering [10]. This places interacting integrable systems in a distinguished position, and raises the question whether interactions induce physically discernible features among equilibrium states. The objective of this paper is to establish a framework to address this.

We focus our study on the isotropic Heisenberg spin-1/2 chain, a paradigmatic model of exactly solvable quantum many-body dynamics, due to both its simplicity and physical relevance. Our main result is an explicit construction of the entire manifold of equilibrium states, which helps expose a rich structure intrinsically linked to inter-particle interactions. We term this the '*equilibrium landscape*'.

Studies of the thermodynamic properties of exactly solvable models have been traditionally focused on canonical Gibbs equilibrium. Only in recent years has interest shifted towards the Generalized Gibbs ensembles, predominantly discussed in the context of quantum quench dynamics in several physically relevant models such as the anisotropic Heisenberg model and Lieb-Liniger Bose gas [4,11–13]. Here prominence was given to simple initial states of potential experimental relevance, while more recent works have considered more generally a special class of 'integrable' product states [14–17]. In the present work we pursue a general and systematic characterisation of the entire space of equilibrium ensembles without appealing to any initial state specific considerations.

Throughout the work we shall employ a range of techniques from the integrability toolbox, combining the algebraic, thermodynamic, and functional Bethe ansatz approaches. We begin by formulating an explicit algebraic construction of the GGE, and then proceed to analyse two complementary routes for evaluating equilibrium partition sums. On the one hand, the celebrated Thermodynamic Bethe Ansatz (TBA) approach [18–20] casts the partition function as a functional integral, invoking a spectral resolution through coupled interacting quasi-particle modes. A saddle-point of the functional integral yields an infinite set of coupled integral equations encoding the equilibrium state. On the other hand, we achieve a regularisation of a general partition function as a two-dimensional classical vertex model where, similarly as in the Gibbs canonical ensemble [21–27], the main subject of study is the dominant eigenvalue of a column transfer matrix. To our knowledge, no previous work on the statistical mechanics of exactly solvable models, including a large body of work on solvable classical vertex models, has achieved a similarly comprehensive description of the entire equilibrium manifold of a model.

For the case of canonical Gibbs equilibrium, compatibility of the two approaches to thermodynamics has been already demonstrated previously in [26]. By means of an integrable Trotterisation of the density operator, usually referred to as the Quantum Transfer Matrix, the Gibbs free energy can be expressed as a solution to the non-linear integral equation [24,25,27,28].

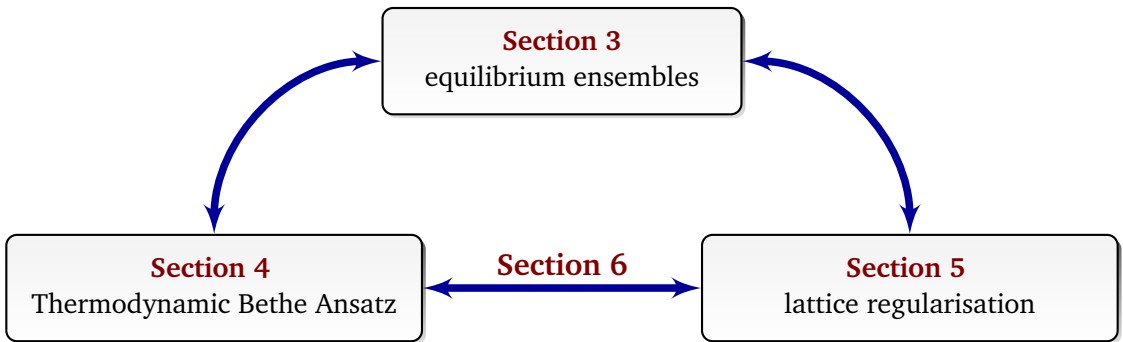

Figure 1: **Outline of the paper.** In Section 2 we introduce the Heisenberg spin-1/2 chain and define the main objects of the integrability framework. In Section 3 we discuss generic equilibrium ensembles and specify the general density matrix. In Section 4 we cover the TBA approach and systematically discuss analytic properties of generic macrostates. In Section 5 we regularise the general density matrix and recast it as a two-dimensional classical vertex model. In Section 6 we define the mirror system and employ functional Bethe ansatz to demonstrate equivalence with TBA.

In this regard, a special and seemingly non-generic analytic structure of the transfer matrix spectrum turns out to be crucial. In this work we demonstrate that typical macrostates from the equilibrium landscape have a much richer structure which necessitate going beyond the conventional Trotterisation techniques. Through achieving this we establish compatibility with the TBA formalism on the general grounds, and find that this yields a clear and instructive picture of the emergence of the equilibrium landscape.

## 2 The Heisenberg spin chain

In this work we outline our construction for perhaps the most widely studied interacting integrable quantum system, the one-dimensional isotropic spin-1/2 Heisenberg model [19, 20, 29, 30]

$$\mathbf{H} = J \sum_{i=1}^{L} \left( \frac{1}{4} - \vec{\mathbf{S}}_i \cdot \vec{\mathbf{S}}_{i+1} \right), \tag{2.1}$$

with exchange coupling $J$ (here $J > 0$ corresponds to a ferromagnetic ground state) and periodic boundary conditions $\vec{\mathbf{S}}_{L+1} = \vec{\mathbf{S}}_1$. The $\vec{\mathbf{S}} \equiv (\mathbf{S}^x, \mathbf{S}^y, \mathbf{S}^z)$ are the local generators of the $\mathfrak{su}(2)$ algebra, $[\mathbf{S}^\alpha, \mathbf{S}^\beta] = i \epsilon_{\alpha\beta\gamma} \mathbf{S}^\gamma$. The model possesses a manifest global $\mathfrak{su}(2)$ symmetry,

$$\left[ \mathbf{H}, \mathbf{S}_{\text{tot}}^\alpha \right] = 0, \tag{2.2}$$

with $\mathbf{S}_{\text{tot}}^\alpha = \sum_{i=1}^{L} \mathbf{S}_i^\alpha$. In this section we summarise the integrable structure of the model, introducing the concepts and notations which form the foundation of the work.

**Spectrum.** The degrees of freedom of the spin chain are magnon excitations, corresponding to spin waves with respect to a reference fully polarised ferromagnetic vacuum. A key feature of integrability is that the magnons undergo non-trivial yet non-diffractive scattering, implying that any interaction process can be reduced to a sequence of two-particle scatterings. With respect to the vacuum state, the quantisation conditions for the magnons are known as the

Bethe equations [29]

$$e^{ik(u_i)L} \prod_{j=1, j\neq i}^{M} S(u_i - u_j) = 1. \tag{2.3}$$

Here the magnons are conveniently parametrised by a rapidity variable $u$, through which their momentum is

$$k(u) = \frac{1}{i} \log\left( \frac{u + \frac{i}{2}}{u - \frac{i}{2}} \right), \tag{2.4}$$

and the two-magnon scattering amplitude is given by

$$S(u) = \frac{u - i}{u + i}, \tag{2.5}$$

which depends only on the difference of rapidities. Due to the $\mathfrak{su}(2)$ symmetry of the model the eigenstates arrange into degenerate $\mathfrak{su}(2)$ multiplets, and each highest-weight state (with respect to the orientation of the reference ferromagnetic vacuum) corresponds to a set of magnon rapidities $\{u_i\}$ satisfying Eqs. (2.3). The corresponding energy eigenvalue has the additive form

$$E = J \sum_{i=1}^{M} \left( 1 - \cos k(u_i) \right). \tag{2.6}$$

The descendant states in a multiplet are obtained by adding zero momentum magnons, i.e. rapidities with $u = \infty$, for which the scattering amplitude trivialises.

**Bound states.** The scattering between magnons induces bound state formation. These correspond to the collections of complex magnon rapidities aligning into 'string' patterns in the complex rapidity plane. In the large-$L$ limit the Bethe roots are classified according to the 'string hypothesis',

$$\bigcup_{i=1}^{M} \{u_i\} \longmapsto \bigcup_{j=1}^{\infty} \bigcup_{i=1}^{M_j} \bigcup_{a=1}^{j} \left\{ u_{j,i} + (j+1-2a)\frac{i}{2} \right\}, \tag{2.7}$$

with all $u_{j,i} \in \mathbb{R}$. Bound states of $j$ magnons are accordingly called $j$-strings, and the set of $j$-strings provide the thermodynamic particle content of the model, i.e. $M = \sum_{j=1}^{\infty} M_j$. The string rapidities are subject to the 'string Bethe equations',

$$e^{ik_j(u_{j,i})L} \prod_{j'=1}^{\infty} \prod_{i'=1}^{M_j} S_{j,j'}(u_{j,i} - u_{j',i'}) = -1, \tag{2.8}$$

valid up to corrections which are suppressed in system size $L$. Here the bare momentum of a $j$-string is

$$k_j(u) = \frac{1}{i} \log\left( \frac{u + j\frac{i}{2}}{u - j\frac{i}{2}} \right), \tag{2.9}$$

and the scattering amplitudes between a $j$-string and a $\ell$-string are

$$S_{j,\ell}(u) = \prod_{a=1}^{j} \prod_{b=1}^{\ell} S\left( u + (j - \ell - 2a + 2b)\frac{i}{2} \right). \tag{2.10}$$

The factor $-1$ on the right-hand side of Eq. (2.8) compensates the self-scattering factor $S_{j,j}(0) = -1$ from the left-hand side.

**Macrostates.** In the thermodynamic limit, defined as $L, M \to \infty$ with ratio $M/L$ kept fixed, the string rapidities distribute densely along the real line. Macrostates are characterised by the complete set of the string rapidity densities $\{\rho_j(u)\}$, with $L\rho_j(u)\mathrm{d}u$ being the number of occupied $j$-string modes in an infinitesimal rapidity interval $\mathrm{d}u$ around $u$. These obey the log-differential form of Eq. (2.8), the Bethe–Yang equations,

$$\rho_j + \bar{\rho}_j = \frac{1}{2\pi}\left|\frac{\mathrm{d}k_j}{\mathrm{d}u}\right| - K_{j,\ell} \star \rho_\ell, \tag{2.11}$$

where $\bar{\rho}_j(u)$ denotes the corresponding density of holes, i.e. unoccupied modes, and the scattering kernels

$$K_{j,\ell}(u) = \frac{1}{2\pi\mathrm{i}}\partial_u \log S_{j,\ell}(u), \tag{2.12}$$

are the differential scattering phases. Here we use the following short-hand notation for matrix convolutions

$$F_{j,\ell} \star f_\ell \equiv \sum_{\ell=1}^{\infty}\int_{-\infty}^{\infty}\mathrm{d}w F_{j,\ell}(u-w)f_\ell(w), \tag{2.13}$$

and adopt the summation convention for the repeated indices. In addition, we also use a short-hand notation for scalar integrations as follows

$$f \star g \equiv \int_{-\infty}^{\infty}\mathrm{d}w f(u-w)g(w), \qquad f \circ g \equiv \int_{-\infty}^{\infty}\mathrm{d}w f(w)g(w). \tag{2.14}$$

For each macrostate there is an associated entropy, which is the logarithm of the number corresponding microstates. This is expressed through the entropy density functional [18, 20]

$$\mathfrak{s}[\rho_j, \bar{\rho}_j] = (\rho_j + \bar{\rho}_j)\log(\rho_j + \bar{\rho}_j) - \rho_j\log\rho_j - \bar{\rho}_j\log\bar{\rho}_j, \tag{2.15}$$

where $\exp\left(L\mathfrak{s}[\rho_j(u), \bar{\rho}_j(u)]\mathrm{d}u\right)$ counts the number of ways of distributing $L\rho_j(u)\mathrm{d}u$ particles between the $L(\rho_j(u) + \bar{\rho}_j(u))\mathrm{d}u$ many $j$-string mode numbers on an infinitesimal rapidity interval $\mathrm{d}u$ centred at $u$.

**Kernel identities.** The scattering kernels $K_{j,\ell}$ are differential scattering phase shifts which encode interactions at the level of macrostates. They exhibit a rich structure which we will exploit throughout this work. Firstly, the Fredholm operator $(1 + K)$ admits a pseudo-inverse $(1 - R)$ through

$$(1-R)_{j,\ell} \star (1+K)_{\ell,k} = 1, \tag{2.16}$$

with $1 \equiv \delta_{j,k}\delta(u)$, where the Fredholm resolvent,

$$R_{j,\ell}(u) = I_{j,\ell}\,s(u), \tag{2.17}$$

is defined through the $s$-kernel

$$s(u) = \frac{1}{2\cosh(\pi u)}, \tag{2.18}$$

and the nearest-neighbour incidence matrix $I$,

$$I_{j,\ell} = \delta_{j-1,\ell} + \delta_{j+1,\ell}. \tag{2.19}$$

Here $(1 - R)$ admits a non-trivial nullspace and is thus not the true inverse of $(1 + K)$. In particular, the relation $(1 - R)_{j,\ell} \star n_\ell = 0$, with boundary condition $n_0 = 0$, has a one-parameter

solution $n_j = hj$. Similarly the $s$-kernel admits a pseudo-inverse $s^{-1}$, which is a left-inverse under convolution, i.e. $s^{-1} \star s \star f = f$, and which also possesses a non-trivial nullspace. Explicitly it is,

$$(s^{-1} \star f)(u) = f(u + \tfrac{i}{2} - i\epsilon) + f(u - \tfrac{i}{2} + i\epsilon),\qquad(2.20)$$

where $\epsilon \equiv 0^+$ is an essential positive infinitesimal which prescribes the avoidance of the poles of $s(u)$ at $u = \pm\tfrac{i}{2}$. The nullspace of $s^{-1}$, generated by functions $\zeta$ obeying $s^{-1} \star \zeta = 0$, is a linear span of basis functions $\log \tau(u; w)$ of the form

$$\log \tau(u; w) \equiv \log \tanh\left(\tfrac{\pi}{2}(u - w)\right), \qquad w \in \mathcal{P},\qquad(2.21)$$

where $\mathcal{P}$ is a strip in the complex plane defined as

$$\mathcal{P} = \left\{ u \in \mathbb{C} : |\mathrm{Im}(u)| \leq \tfrac{1}{2} - \epsilon \right\},\qquad(2.22)$$

commonly referred to as the 'physical strip'. We highlight the explicit dependence on the infinitesimal regulator $\epsilon$ here, which ensures that the boundaries $\mathrm{Im}(u) = \tfrac{1}{2}$ are excluded from the strip, as it will prove useful in later sections. The functions $\log \tau(u; w)$ are related back to the $s$-kernel through the identity

$$s(u) = \mp \frac{1}{2\pi i} \partial_u \log \tau(u; \pm \tfrac{i}{2}).\qquad(2.23)$$

A related object is the discrete d'Alembertian operator

$$\Box = s^{-1} \star (1 - R) = s^{-1} - I,\qquad(2.24)$$

or explicitly,

$$(\Box f)_j(u) = f_j(u + \tfrac{i}{2} - i\epsilon) + f_j(u - \tfrac{i}{2} + i\epsilon) - f_{j-1}(u) - f_{j+1}(u),\qquad(2.25)$$

for a set of functions $f_j(u)$. The associated Green's function, obeying $\Box G = 1$, is given by

$$G_{j,k} = (1 + K)_{j,k} \star s.\qquad(2.26)$$

The d'Alembertian $\Box$ inherits a non-trivial nullspace from both $(1 - R)$ and $s^{-1}$. From $(1 - R)$ the functions $n_j = hj$ obey $\Box n = 0$, while given a set of functions $\zeta_j$ in the nullspace of $s^{-1}$, there exist related functions $\nu_j = (1 + K)_{j,\ell} \star \zeta_\ell$ which also satisfy $\Box \nu = 0$. It is further useful to define kernels $K_j$ and their associated amplitudes $S_j$ as follows

$$K_j(u) = \frac{1}{2\pi i} \partial_u \log S_j(u) = \frac{1}{2\pi i}\left(\frac{1}{u - j\tfrac{i}{2}} - \frac{1}{u + j\tfrac{i}{2}}\right), \qquad S_j(u) = \frac{u - j\tfrac{i}{2}}{u + j\tfrac{i}{2}},\qquad(2.27)$$

with the kernels obeying the identities

$$\frac{1}{2\pi}\left|\frac{dk_j}{du}\right| = K_j(u), \qquad (1 - R)_{j,\ell} \star K_\ell = \delta_{j,1} s.\qquad(2.28)$$

These provide convenient explicit expressions for the matrix elements of the Green's function $G_{j,k}$ and its associated amplitude $\Psi_{j,k}$ as follows

$$G_{j,k}(u) = \frac{1}{2\pi i} \partial_u \log \Psi_{j,k}(u) = \sum_{a=1}^{\min(j,k)} K_{j+k+1-2a}(u), \qquad \Psi_{j,k}(u) = \prod_{a=1}^{\min(j,k)} S_{j+k+1-2a}(u),\qquad(2.29)$$

and in turn for the scattering kernels and their amplitudes through

$$K_{j,k}(u) = I_{j,\ell} G_{\ell,k}(u), \qquad \log S_{j,k}(u) = I_{j,\ell} \log \Psi_{\ell,k}(u).\qquad(2.30)$$

Finally, we emphasise that the $\epsilon$ regulator is tied to the pseudo-inverse $s^{-1}$, and in particular does not appear in the string compounds in Eq. (2.7), nor in the general definitions of kernels and amplitudes, e.g. Eqs. (2.10), (2.27). In the following we employ a compact notation for half-unit imaginary shifts not involving a regulator

$$f^{\pm}(v) = f(v \pm \tfrac{i}{2}).\qquad(2.31)$$

**Transfer matrices.** The algebraic formulation of integrability is founded upon the Lax representation [31–33], see also [34–37] and references therein. The Lax matrices are a family of operators $\mathbf{L}_{k,1}(v, u) : \mathcal{V}_k \otimes \mathcal{V}_1 \to \mathcal{V}_k \otimes \mathcal{V}_1$, where $\mathcal{V}_k$ denotes the $(k+1)$-dimensional irreducible unitary representation of $\mathfrak{su}(2)$, of the form

$$\mathbf{L}_{k,1}(v, u) = (v - u)\mathbf{1} \otimes \mathbf{1} + 2\mathrm{i} \sum_{\alpha=\mathrm{x},\mathrm{y},\mathrm{z}} \mathbf{S}^\alpha \otimes \mathbf{S}^\alpha. \tag{2.32}$$

These provide local building blocks for the transfer matrices, a commuting family of operators acting on the Hilbert space $\mathcal{H} \cong \mathcal{V}_1^{\otimes L}$, which are given as traces over path-ordered products of operators $\mathbf{L}_{k,1}$,

$$\mathbf{T}_k(v) = \mathrm{Tr}_{\mathcal{V}_k} \mathbf{L}_{k,1}(v, 0) \otimes \mathbf{L}_{k,1}(v, 0) \otimes \cdots \otimes \mathbf{L}_{k,1}(v, 0), \tag{2.33}$$

where the trace is taken over the common auxiliary space $\mathcal{V}_k$, with $k \in \mathbb{N}$. Trivially, $\mathbf{T}_0(v) = v^L$. Integrability ensures that the transfer matrices $\mathbf{T}_k(v)$ mutually commute,

$$\left[ \mathbf{T}_k(v), \mathbf{T}_{k'}(v') \right] = 0, \tag{2.34}$$

for all values of $k, k' \in \mathbb{N}$ and $v, v' \in \mathbb{C}$.

**Fusion hierarchy.** The eigenvalues $T_k(v)$ of the transfer matrices $\mathbf{T}_k(v)$ are called $T$-functions. They are polynomial and satisfy the Hirota equation[1]

$$T_k^+(v)T_k^-(v) = \phi_k(v)\bar{\phi}_k(v) + T_{k-1}(v)T_{k+1}(v), \qquad k \geq 0, \tag{2.35}$$

with initial conditions $T_{-1} \equiv 0$, $T_0(v) = v^L$, and boundary 'scalar potentials',

$$\phi_k(v) = \left(v + (k+1)\tfrac{\mathrm{i}}{2}\right)^L, \quad \bar{\phi}_k(v) = \left(v - (k+1)\tfrac{\mathrm{i}}{2}\right)^L. \tag{2.36}$$

The Hirota equation exhibits a gauge freedom corresponding to the overall normalisation of the Lax matrix. There however exist $Y$-functions,

$$Y_k(v) = \frac{T_{k-1}(v)T_{k+1}(v)}{\phi_k(v)\bar{\phi}_k(v)} = \frac{T_k^+(v)T_k^-(v)}{\phi_k(v)\bar{\phi}_k(v)} - 1, \tag{2.37}$$

which are gauge-invariant quantities satisfying the *canonical Y-system hierarchy* [41, 42]

$$Y_k^+(v)Y_k^-(v) = \left(1 + Y_{k-1}(v)\right)\left(1 + Y_{k+1}(v)\right), \qquad k \geq 1, \tag{2.38}$$

with initial condition $Y_0(v) = 0$.

**Thermodynamic inversion identity.** In the large-$L$ limit, the entire family of transfer matrices $\mathbf{T}_k(v)$ satisfy the useful identity [43]

$$\lim_{L\to\infty} \frac{\mathbf{T}_k^+(v)\mathbf{T}_k^-(v)}{\phi_k(v)\bar{\phi}_k(v)} = \mathbf{1}, \tag{2.39}$$

which allows for their inversion. This property can equivalently be expressed as the large-$L$ decay of the physical $Y$-functions

$$\lim_{L\to\infty} Y_k(v) = 0, \qquad v \in \mathcal{P}. \tag{2.40}$$

---

[1]The Hirota equation (2.35) can be understood as the 'quantum' counterpart of the fusion identities for characters of the 'classical' Lie algebra $\mathfrak{su}(2)$, i.e. fusion identities amongst unitary irreducible representations of $\mathfrak{su}(2)$. Additional details on fusion identities can be found in e.g. [24, 34, 35, 38–40].

**Local charges.** The transfer matrices serve as generating operators for the local charges through their logarithmic derivatives [4,43],

$$\mathbf{X}_k(\nu) = \frac{1}{2\pi i} \partial_\nu \log \frac{\mathbf{T}_k^+(\nu)}{\phi_k(\nu)}. \tag{2.41}$$

These charges are well-defined only on the physical strip, $\nu \in \mathcal{P}$, specified above in Eq. (2.22). When $\nu$ approaches the boundary of the strip, the charges acquire a divergent localisation length [43,44] and thus become singular at the boundaries $\partial\mathcal{P} = \mathrm{Im}(\nu) = \pm\frac{1}{2}$.

The Hamiltonian Eq. (2.1) is given by $\mathbf{H} = \pi J\,\mathbf{X}_1(0)$.

**String charge-duality.** In the thermodynamic limit, the eigenvalues of the charges $\mathbf{X}_k(\nu)$ are expressible as a linear functional of the rapidity densities [7]

$$X_k = G_{k,j} \star \rho_j, \tag{2.42}$$

where $G$, given in Eq. (2.26), is the Green's function of the d'Alembertian $\Box$. The inverted relation, promoted to the level of operators,

$$\boldsymbol{\rho} = \Box\,\mathbf{X}, \tag{2.43}$$

bears the name 'string-charge duality'. We highlight that this defines the mode operators $\boldsymbol{\rho}_j$ in a manner which is independent of the of the orientation of the reference ferromagnetic vacuum.

Even though the positive infinitesimal $\epsilon$ does not appear in the definition of the charges $\mathbf{X}_k(\nu)$, the mode operators $\boldsymbol{\rho}_j$ inherit the $\epsilon$-prescription through the left-inverse $s^{-1}$ which enters in the d'Alembertian $\Box$, Eq. (2.25). The important consequence of the regulator $\epsilon$ is that the boundaries $\partial\mathcal{P}$ at $\mathrm{Im}(u) = \frac{1}{2}$ are avoided, ensuring a finite localisation length of the $\boldsymbol{\rho}_j$, i.e. as $\epsilon$ is strictly positive the localisation lengths are strictly finite, cf. [43,44]. Thus $\epsilon$ admits a physical interpretation as a regulator which governs the notion of locality in the large-$L$ limit.

# 3 Equilibrium ensembles

The purpose of this article is to characterise the equilibrium landscape of an interacting integrable model, specifically the Heisenberg spin chain. Equilibrium ensembles emerge dynamically in the long-time limit of unitary evolution from generic initial states in thermodynamically large systems. These can be viewed equivalently in either the canonical sense as density matrices governing the expectation values of all local observables, or in the microcanonical sense as unbiased collections of eigenstates sharing the same values of all local charge densities. There are two key mechanisms underlying local equilibration: (i) decoherence, causing the dynamical phases between individual eigenstates to average out during the relaxation process, and (ii) the eigenstate thermalisation hypothesis which supposes the local equivalence of distinct eigenstates from the same microcanonical shell [1–3,45]. Local correlation function are thus expressible as functionals of the quasi-particle densities characterising a macrostate, see e.g. [46–48].

For the Heisenberg spin chain, a microcanonical shell corresponds to the set of microstates associated with a given macrostate, parametrised by the full set of occupied mode distributions $\rho_j(u)$. In this context, the statement of the eigenstate thermalisation hypothesis is substantiated by the Yang–Yang form of the entropy density, Eq. (2.15), see e.g. [6,49,50]. The corresponding (unnormalised) density matrix is [51]

$$\varrho = \exp\Big[-\mu_j \circ \boldsymbol{\rho}_j + \vec{h}\cdot\vec{\mathbf{S}}_{\mathrm{tot}}\Big], \tag{3.1}$$

where $\mu_j(u)$ provide a general set of chemical potentials for the mode operators $\boldsymbol{\rho}_j$. The term $\vec{h} \cdot \vec{S}_{\text{tot}}$ incorporates a general Cartan charge of the global $\mathfrak{su}(2)$ symmetry, which serves to specify the polarisation direction $\vec{h} = (h_x, h_y, h_z)$ with respect to which the Bethe magnons are defined, i.e. with respect to a ferromagnetic vacuum oriented in the $\vec{h}$ direction. For consistency, the chemical potentials must not diverge with $j$, i.e.

$$\lim_{j \to \infty} \frac{\mu_j(u)}{j} = 0. \tag{3.2}$$

The above functional parametrisation of the density matrix differs from the more commonly used definition involving a formal infinite sum over a discrete basis of local conservation laws (see e.g. [4–6,52]) or a 'truncated' GGE [53–55]. We argue however that gauge-invariant formulation (3.1) not only clearly conveys the physical picture underlying the GGE concept, it moreover provides a natural and convenient starting point for analysis of the ensemble as developed in the following sections.

## 4 Thermodynamic Bethe Ansatz

In this section we revisit the formalism of Thermodynamic Bethe Ansatz, a functional integral formulation of thermodynamics [18–20]. Partition sums are cast in the basis of Bethe eigenstates, which in the thermodynamic limit translates to functional integration in the space of macrostates. In particular, the (generalized) free energy density,

$$f = -\lim_{L \to \infty} \frac{1}{L} \log \text{Tr}_{\mathcal{H}} \boldsymbol{\varrho}, \tag{4.1}$$

is reformulated as a functional integral over the string rapidity distributions,

$$f = \int \mathcal{D}[\{\rho_j\}] \Big( (\mu_j + hj) \circ \rho_j - \sum_j \mathfrak{s}[\rho_j, \bar{\rho}_j] \Big), \tag{4.2}$$

with $h = |\vec{h}|$, and the entropy functional $\mathfrak{s}$ is specified by Eq. (2.15). Identifying the saddle point of the equilibrium partition sum through $\delta f / \delta \rho_j(u) = 0$, subject to the constraints (2.11), yields the celebrated TBA equations,

$$\log \mathscr{Y}_j = \mu_j + hj + K_{j,\ell} \star \log\big(1 + 1/\mathscr{Y}_\ell\big), \tag{4.3}$$

expressed in terms of the 'thermodynamic $\mathscr{Y}$-functions',

$$\mathscr{Y}_j(u) = \frac{\bar{\rho}_j(u)}{\rho_j(u)}. \tag{4.4}$$

The equilibrium free energy density then becomes

$$f = -\frac{1}{2\pi} \left| \frac{\mathrm{d}k_j}{\mathrm{d}u} \right| \circ \log(1 + 1/\mathscr{Y}_j), \tag{4.5}$$

which can be equivalently expressed in the form

$$f = s \circ \mu_1 - s \circ \log(1 + \mathscr{Y}_1), \tag{4.6}$$

obtained by inserting Eq. (2.16) in Eq. (4.5), and making use of the identities Eq. (2.28), the TBA equations Eq. (4.3), and that $hj$ belongs to the nullspace of $(1 - R)$.

The TBA equations provide the link between the chemical potentials $\mu_j$ and the thermodynamic functions of general equilibrium states. To further elucidate the underlying structure, we next bring the TBA equations to a local form by convolving with the left-inverse $(1-R)$ of the Fredholm operator $(1+K)$, resulting in

$$\log \mathscr{Y}_j = d_j + I_{j,\ell}\, s \star \log(1 + \mathscr{Y}_\ell), \tag{4.7}$$

with $\mathscr{Y}_0 \equiv 0$, and source terms

$$d_j = (1-R)_{j,\ell} \star \mu_\ell. \tag{4.8}$$

Here information about $h$, which has dropped from the source term as it belongs to the nullspace of $(1-R)$, instead appears implicitly through the large-$j$ asymptotics

$$\lim_{j \to \infty} \frac{\log \mathscr{Y}_j(u)}{j} = h, \tag{4.9}$$

which must be supplemented to Eqs. (4.7) in order to unambiguously fix a solution. The information stored in $\mu_j$ is preserved by $(1-R)$, as condition Eq. (3.2) forbids a nullspace contribution, and so Eq. (4.8) can be readily inverted

$$\mu_j = (1+K)_{j,\ell} \star d_\ell. \tag{4.10}$$

The TBA equations are integral equations defined on the real rapidity axis. We now analytically continue them to the complex rapidity plane, obtaining an equivalent set of functional relations called the 'thermodynamic $\mathscr{Y}$-system' [51]. This is achieved by convolving both sides of Eqs. (4.7) with the pseudo-inverse $s^{-1}$ and subsequently exponentiating, resulting in

$$\mathscr{Y}_j^+(u - \mathrm{i}\epsilon)\mathscr{Y}_j^-(u + \mathrm{i}\epsilon) = e^{\lambda_j(u)}\big(1 + \mathscr{Y}_{j-1}(u)\big)\big(1 + \mathscr{Y}_{j+1}(u)\big), \tag{4.11}$$

with

$$\lambda_j = s^{-1} \star d_j. \tag{4.12}$$

This provides a natural decomposition of the source terms,

$$d_j = s \star \lambda_j + \zeta_j, \tag{4.13}$$

where $\zeta_j$ satisfy

$$s^{-1} \star \zeta_j = 0. \tag{4.14}$$

Owing to Eq. (2.21), we adopt the following generic parametrisation

$$\zeta_j(u) = \sum_i \alpha_{j,i} \log[\tau(u; w_{j,i})\tau(u; \bar{w}_{j,i})], \tag{4.15}$$

in terms of parameters $\alpha_{j,i} \in \mathbb{R}$, and complex-conjugate $w_{j,i}$ and $\bar{w}_{j,i}$ located in $\mathcal{P}$. Although the sum involves a finite number of terms, infinite convergent sequences of simple poles and zeros ($\alpha_{j,i} = \pm 1$) which condense along certain contours are also permissible and can be understood as limits of Padé approximants for complex functions with branch points. We adopt the convention that all branch cuts in $\mathcal{P}$ extend vertically away from the real axis.

The decomposition Eq. (4.13) can also be expressed at the level of the chemical potentials

$$\mu_j = G_{j,\ell} \star \lambda_\ell + \nu_j, \tag{4.16}$$

where here $\lambda = \Box \mu$, and $\nu_j$ encodes the nullspace of $\mu_j$ inherited from $s^{-1}$ through

$$\nu_j = (1+K)_{j,\ell} \star \zeta_\ell. \tag{4.17}$$

Indeed the thermodynamic $\mathscr{Y}$-system can be obtained directly from the TBA equations Eq. (4.3) by applying the d'Alembertian $\square$ and exponentiating.

To this point the analysis appears completely formal. It however reveals a structure in the space of equilibrium states. The information from the nullspace of $s^{-1}$, Eq. (2.21), which appears to have dropped out from Eq. (4.11), enters instead implicitly via analytic properties of functions $\mathscr{Y}_j$. Specifically, $w_{j,i}$ and $\bar{w}_{j,i}$ are nothing but branch points of $\mathscr{Y}_j$ of degree $\alpha_{j,i}$. To establish this, we now re-obtain Eq. (4.7) from Eq. (4.11). In order to undo the complex shifts and transform Eq. (4.11) to the real axis, we introduce adapted $\mathscr{Y}$-functions,

$$\mathscr{Y}_j^{\mathrm{adp}}(u) = \mathscr{Y}_j(u) \prod_i \big(\tau(u; w_{j,i}) \tau(u; \bar{w}_{j,i})\big)^{-\alpha_{j,i}}, \tag{4.18}$$

which obey $\mathscr{Y}_j^{\mathrm{adp}}(u + \tfrac{\mathrm{i}}{2} - \mathrm{i}\epsilon) \mathscr{Y}_j^{\mathrm{adp}}(u - \tfrac{\mathrm{i}}{2} + \mathrm{i}\epsilon) = \mathscr{Y}_j(u + \tfrac{\mathrm{i}}{2} - \mathrm{i}\epsilon) \mathscr{Y}_j(u - \tfrac{\mathrm{i}}{2} + \mathrm{i}\epsilon)$ due to $\tau^+ \tau^- = 1$, and are (by construction) analytic on $\mathcal{P}$ with constant large-$u$ asymptotics. Substituting these into Eq. (4.11), taking the logarithm and convolving with $s$ we readily recover Eq. (4.7). The analytic properties of the thermodynamic $\mathscr{Y}$-functions in $\mathcal{P}$ are therefore completely determined by the data $w_{j,i}$ and $\bar{w}_{j,i}$ and $\alpha_{j,i}$. If all $\alpha_{j,i} \in \mathbb{Z}$ then the $\mathscr{Y}$-functions are meromorphic on $\mathcal{P}$, while more generally they possess non-integer branch points.

## 5 Integrable lattice regularisation

In this section we derive an explicit lattice regularisation of a generic equilibrium ensemble. This is achieved this by re-expressing the density matrix as a product of transfer matrices, thereby casting it as a two-dimensional classical vertex model on a cylinder.

The mode operators $\boldsymbol{\rho}_j$ are related to the transfer matrices via string-charge duality, Eq. (2.43). The density matrix Eq. (3.1) can thus be presented in the form

$$\boldsymbol{\varrho} = \exp\Big[ -\mu_j \circ (\square \mathbf{X})_j + \vec{h} \cdot \vec{\mathbf{S}}_{\mathrm{tot}} \Big], \tag{5.1}$$

where the charges $\mathbf{X}_j$ are the logarithmic derivatives of the transfer matrices, Eq. (2.41). To proceed it is necessary to be careful about the potential nullspace of $\mu_j$ under action of the d'Alembertian $\square$. As described in Section 2, this nullspace is spanned by functions $n_j$ in the nullspace of $(1 - R)$, along with functions $\zeta_j$ in the nullspace of $s^{-1}$. The condition Eq. (3.2) forbids a contribution $n_j$, and so the chemical potential decomposes as

$$\mu_j = G_{j,\ell} \star \lambda_\ell + \nu_j, \tag{5.2}$$

where $\lambda = \square \mu$, and $\nu_j = (1 + K)_{j,\ell} \star \zeta_\ell$ encode the nullspace inherited from $s^{-1}$, $\square \nu = 0$. This essentially repeats the derivation of Eq. (4.16) in Section 4. In the following we will adopt the same generic parametrisation of $\zeta_j$ as in Eq. (4.15).

The decomposition of $\mu_j$ induces a factorisation of the density matrix Eq. (3.1) into three commuting factors

$$\boldsymbol{\varrho} = \boldsymbol{\varrho}_\lambda \cdot \boldsymbol{\varrho}_\nu \cdot \boldsymbol{\varrho}_{\vec{h}}, \tag{5.3}$$

given respectively by

$$\boldsymbol{\varrho}_\lambda = \exp\Big[ -\lambda_k \circ \mathbf{X}_k \Big], \qquad \boldsymbol{\varrho}_\nu = \exp\Big[ -\nu_j \circ \boldsymbol{\rho}_j \Big], \qquad \boldsymbol{\varrho}_{\vec{h}} = \exp\Big[ \vec{h} \cdot \vec{\mathbf{S}}_{\mathrm{tot}} \Big], \tag{5.4}$$

where the first factor is obtained through $(G \star \lambda) \circ (\square \mathbf{X}) = (\square G \star \lambda) \circ \mathbf{X} = \lambda \circ \mathbf{X}$. For convenience we will refer to $\boldsymbol{\varrho}_\lambda$ as encoding the 'node data', $\boldsymbol{\varrho}_\nu$ as encoding the 'analytic data', and $\boldsymbol{\varrho}_{\vec{h}}$ as encoding the Cartan charge. We now proceed to analyse each factor in turn.

**The node data.**   The first factor of the ensemble encodes the node data $\lambda_k$,

$$\varrho_\lambda = \exp\left[-\sum_k \lambda_k \circ \mathbf{X}_k\right]. \tag{5.5}$$

To begin we regularise the $\lambda_k$, by invoking a large-rapidity cut-off $\Lambda^\infty$, and casting $\lambda_k(u)$ as a discrete sum on the remaining interval. First it is useful to introduce, for each $k$ separately, two disjoint domains $\{-\Lambda^\infty, \Lambda^\infty\} = \mathcal{I}_k^{(+)} \cup \mathcal{I}_k^{(-)}$, such that $\lambda_k(v) \geq 0$ on $\mathcal{I}_k^{(+)}$ and $\lambda_k(v) < 0$ on $\mathcal{I}_k^{(-)}$. That is, we decompose the $\lambda_k$ as

$$\lambda_k(v) = \lambda_k^{(+)}(v) + \lambda_k^{(-)}(v), \tag{5.6}$$

where $\lambda_k^{(\pm)}(v) = 0$ on $\mathcal{I}_k^{(\mp)}$. Each part can then be regularised via a discrete sum of Dirac $\delta$-distributions,

$$\lambda_k^{(\pm)}(v) = \lim_{n_k \to \infty} \frac{\Lambda_k^{(\pm)}}{n_k^{(\pm)}} \sum_{i=1}^{n_k} \delta\big(v - x_{k,i}^{(\pm)}\big), \tag{5.7}$$

where $n_k^{(\pm)} \in \mathbb{N}$ control the resolution of the discretisation, the parameters $x_{k,i}^{(\pm)} \in \mathbb{R}$ are chosen uniformly from the distributions $\lambda_k^{(\pm)}(v)$, and the overall normalisation is fixed by

$$\Lambda_k^{(\pm)} = \int_{\mathcal{I}_k^{(\pm)}} \mathrm{d}v\, \lambda_k^{(\pm)}(v). \tag{5.8}$$

This readily translates to a further factorisation $\varrho_\lambda = \varrho_\lambda^{(+)} \cdot \varrho_\lambda^{(-)}$, with

$$\varrho_\lambda^{(\pm)} = \prod_k \prod_{i=1}^{n_k^{(\pm)}} \exp\left[-\frac{\Lambda_k^{(\pm)}}{n_k^{(\pm)}} \mathbf{X}_k(x_{k,i}^{(\pm)})\right]. \tag{5.9}$$

Referring back to the definition of the charges Eq. (2.41), each exponential factor is written as

$$\mathbf{X}_k(v) = \lim_{\Delta v \to 0} \frac{1}{\Delta v} \frac{1}{2\pi \mathrm{i}} \left[\log \frac{\mathbf{T}_k^+\big(v + \frac{\Delta v}{2}\big)}{\phi_k(v + \frac{\Delta v}{2})} - \log \frac{\mathbf{T}_k^+(v - \frac{\Delta v}{2})}{\phi_k(v - \frac{\Delta v}{2})}\right]. \tag{5.10}$$

Using the inversion identity Eq. (2.39), which is exact in the large-$L$ limit, this becomes

$$\mathbf{X}_k(v) = \lim_{\Delta v \to 0} \frac{1}{\Delta v} \frac{1}{2\pi \mathrm{i}} \log\left[\frac{\mathbf{T}_k^+\big(v + \frac{\Delta v}{2}\big) \mathbf{T}_k^-(v - \frac{\Delta v}{2})}{\phi_k(v + \frac{\Delta v}{2}) \bar{\phi}_k(v - \frac{\Delta v}{2})}\right]. \tag{5.11}$$

Now, for each factor in Eq. (5.9) we couple the finite difference to the corresponding discretisation parameters $n_k^{(\pm)}$ through the identifications $v \to x_{k,i}^{(\pm)}$ and $\Delta v \to \mathrm{i}\, \xi_k^{(\pm)}$, with parameters

$$\xi_k^{(\pm)} = \frac{\Lambda_k^{(\pm)}}{2\pi\, n_k^{(\pm)}}, \tag{5.12}$$

resulting in

$$\varrho_\lambda^{(\pm)} = \prod_k \prod_{i=1}^{n_k^{(\pm)}} \frac{\mathbf{T}_k^+\big(x_{k,i}^{(\pm)} + \xi_k^{(\pm)} \frac{\mathrm{i}}{2}\big) \mathbf{T}_k^-\big(x_{k,i}^{(\pm)} - \xi_k^{(\pm)} \frac{\mathrm{i}}{2}\big)}{\phi_k\big(x_{k,i}^{(\pm)} + \xi_k^{(\pm)} \frac{\mathrm{i}}{2}\big) \bar{\phi}_k\big(x_{k,i}^{(\pm)} - \xi_k^{(\pm)} \frac{\mathrm{i}}{2}\big)}. \tag{5.13}$$

Let us highlight that while Eq. (5.13) may in a sense be viewed as an 'integrable Trotterisation', the Suzuki–Trotter formula [56, 57] has not been invoked. Instead, we coupled the resolution of the discretisation of $\lambda_k(u)$ in Eq. (5.7) with the finite-difference approximation $\Delta v$ of the derivative in the definition of the charges.

**The analytic data.** We next analyse the factor

$$\varrho_\nu = \exp\Big[-\sum_j \nu_j \circ \boldsymbol{\rho}_j\Big], \tag{5.14}$$

where $\nu = (1 + K) \star \zeta$, with $\zeta_j$ given generically by Eq. (4.15). Here we substitute $\boldsymbol{\rho} = \Box \mathbf{X}$, bringing it to the form

$$\varrho_\nu = \prod_{j,k} \exp\Big[\int_\mathbb{R} \mathrm{d}z\, \nu_j(z) I_{j,k} \mathbf{X}_k(z) - \int_\mathbb{R} \mathrm{d}z\, \nu_j(z)\delta_{jk}\big(\mathbf{X}_k(z + \tfrac{i}{2} - i\epsilon) + \mathbf{X}_k(z - \tfrac{i}{2} + i\epsilon)\big)\Big]. \tag{5.15}$$

Then exploiting the null property $\Box \nu = 0$, the exponent becomes recast as a sum of two contour integrals

$$\varrho_\nu = \prod_{j,k} \exp\Big[\oint_{\mathcal{C}_+} \mathrm{d}z\, \nu_j^-(z + i\epsilon)\delta_{j,k}\mathbf{X}_k(z) - \oint_{\mathcal{C}_-} \mathrm{d}z\, \nu_j^+(z - i\epsilon)\delta_{j,k}\mathbf{X}_k(z)\Big], \tag{5.16}$$

where contours $\mathcal{C}_+$ and $\mathcal{C}_-$ encircle the upper and lower half of the physics strip $\mathcal{P}$ in the counter-clockwise direction, respectively. Here we have shifted the integration contours using

$$\int_\mathbb{R} \mathrm{d}z\, f(z)\mathbf{X}_k^\pm(z \mp i\epsilon) = \int_\mathbb{R} \mathrm{d}z\, f^\mp(z \pm i\epsilon)\mathbf{X}_k(z) \mp \oint_{\mathcal{C}_\pm} \mathrm{d}z\, f^\mp(z \pm i\epsilon)\mathbf{X}_k(z), \tag{5.17}$$

along with the asymptotic behaviour $\lim_{|u|\to\infty} \nu_j(u) = 0$. Introducing the functions

$$\gamma_j(u) = -\frac{1}{2\pi i}\frac{\mathrm{d}\nu_j(u)}{\mathrm{d}u}, \tag{5.18}$$

and integrating by parts, we obtain

$$\varrho_\nu = \prod_{j,k} \exp\Big[\oint_{\mathcal{C}_+} \mathrm{d}z\, \gamma_j^-(z + i\epsilon)\delta_{jk} \log\frac{\mathbf{T}_k^+(z)}{\phi_k(z)} - \oint_{\mathcal{C}_-} \mathrm{d}z\, \gamma_j^+(z - i\epsilon)\delta_{jk} \log\frac{\mathbf{T}_k^+(z)}{\phi_k(z)}\Big]. \tag{5.19}$$

Specifically, the functions $\gamma_j$ are given through

$$\gamma_j^\pm(u) = \pm\alpha_{j,i} \sum_k \sum_{i=1} \big(G_{j,k}(u - w_{k,i}) + G_{j,k}(u - \bar{w}_{k,i})\big), \tag{5.20}$$

using identities (2.23) and (2.26).

Now we evaluate the above contour integrals. Recalling the explicit expression for $G$ from Eq. (2.29), and noting that the poles of kernels $K_j(u)$ are of the form, $2\pi i \operatorname{Res}_{u=0} K_j(u \pm \tfrac{i}{2}k) = \pm\delta_{jk}$, we observe that the poles of $\gamma_j^\pm(u)$ with residues $\pm\alpha_{j,i}$ are located at the null components shifted by integer multiples of $\tfrac{i}{2}$. Given that all the null components lie within the physical strip $\mathcal{P}$, the only contribution which does not shift the poles outside the contours then comes from the term $K_1(u)$ which appears only in $G_{j,j}(u)$. In effect, the analytic data gets shifted by $\pm\tfrac{i}{2}$, giving a pole in $\mathcal{C}_-$ ($\mathcal{C}_+$) for each null component in $\mathcal{C}_+$ ($\mathcal{C}_-$). Thus Eq. (5.19) simplifies to

$$\varrho_\nu = \prod_k \prod_i \exp\Big[\alpha_{k,i} \log\Big(\frac{\mathbf{T}_k^+(w_{k,i} + i\epsilon - \tfrac{i}{2})}{\phi_k(w_{k,i} + i\epsilon - \tfrac{i}{2})}\frac{\mathbf{T}_k^-(\bar{w}_{k,i} - i\epsilon + \tfrac{i}{2})}{\bar{\phi}_k(\bar{w}_{k,i} - i\epsilon + \tfrac{i}{2})}\Big)\Big], \tag{5.21}$$

upon using the inversion identity Eq. (2.39).

Finally, Eq. (5.21) must be further regularised in order to convert to a product of transfer matrices. To achieve this we note the following property: any term $\alpha \log[\tau(u;w)\tau(u;\bar{w})]$ with $0 < \alpha < 1$ and $w, \bar{w} \in \mathcal{P}$ can be systematically approximated by a sum

$$\sum_a \log\big[\tau(u;w_a^{\mathrm{z}})\,\tau(u;\bar{w}_a^{\mathrm{z}})\big] - \sum_b \log\big[\tau(u;w_b^{\mathrm{p}})\,\tau(u;\bar{w}_b^{\mathrm{p}})\big], \tag{5.22}$$

where the sets of zeros $\{\omega_a^{\mathrm{z}}, \bar{\omega}_a^{\mathrm{z}}\}$ and poles $\{\omega_b^{\mathrm{p}}, \bar{\omega}_b^{\mathrm{p}}\}$ are obtained as the zeros and poles lying within the physical strip $\mathcal{P}$ of a Padé approximation of $(u-w)^\alpha(u-\bar{w})^\alpha$ about $w_0 = \mathrm{Re}(w) = \mathrm{Re}(\bar{w})$. Applying this to each contribution of $\zeta_j$ individually we obtain a regularised form

$$\zeta_j^{\mathrm{reg}}(u) = \sum_{a=1}^{n_j^{\mathrm{z}}} \log\big[\tau(u;\omega_{j,a}^{\mathrm{z}})\,\tau(u;\bar{\omega}_{j,a}^{\mathrm{z}})\big] - \sum_{b=1}^{n_j^{\mathrm{p}}} \log\big[\tau(u;\omega_{j,b}^{\mathrm{p}})\,\tau(u;\bar{\omega}_{j,b}^{\mathrm{p}})\big], \tag{5.23}$$

where both the distinction between distinct zero modes and dependence on the degree of the Padé approximation are left implicit. In this way we obtain the factor $\varrho_\nu$ of the ensemble as a product of transfer matrices

$$\begin{aligned}
\varrho_\nu = \prod_k \prod_{a=1}^{n_k^{\mathrm{z}}} &\frac{\mathbf{T}_k^+(\omega_{k,a}^{\mathrm{z}} + \mathrm{i}\epsilon - \frac{\mathrm{i}}{2})}{\phi_k(\omega_{k,a}^{\mathrm{z}} + \mathrm{i}\epsilon - \frac{\mathrm{i}}{2})} \frac{\mathbf{T}_k^-(\bar{\omega}_{k,a}^{\mathrm{z}} - \mathrm{i}\epsilon + \frac{\mathrm{i}}{2})}{\bar{\phi}_k(\bar{\omega}_{k,a}^{\mathrm{z}} - \mathrm{i}\epsilon + \frac{\mathrm{i}}{2})} \\
\times \prod_k \prod_{b=1}^{n_p^{\mathrm{z}}} &\frac{\mathbf{T}_k^+(\bar{\omega}_{k,b}^{\mathrm{p}} - \mathrm{i}\epsilon + \frac{\mathrm{i}}{2})}{\phi_k(\bar{\omega}_{k,b}^{\mathrm{p}} - \mathrm{i}\epsilon + \frac{\mathrm{i}}{2})} \frac{\mathbf{T}_k^-(\omega_{k,b}^{\mathrm{p}} + \mathrm{i}\epsilon - \frac{\mathrm{i}}{2})}{\bar{\phi}_k(\omega_{k,b}^{\mathrm{p}} + \mathrm{i}\epsilon - \frac{\mathrm{i}}{2})}.
\end{aligned} \tag{5.24}$$

**The Cartan charge.** The final factor of the ensemble is simply given by

$$\varrho_{\vec{h}} = \exp\big[\vec{h} \cdot \vec{\mathbf{S}}_{\mathrm{tot}}\big] \equiv \bigotimes_{j=1}^L \mathbf{D}(\vec{h}), \qquad \mathbf{D}(\vec{h}) = \exp\big[\vec{h} \cdot \vec{\mathbf{S}}\big]. \tag{5.25}$$

There is no trace over an auxiliary space associated with this factor.

## 5.1 The two-dimensional vertex model

Now recombining the three factors the ensemble takes the compact form

$$\varrho = \varrho_{\vec{h}} \prod_k \prod_{i=1}^{N_k} \frac{\mathbf{T}_k^+(\theta_{k,i})}{\phi_k(\theta_{k,i})} \frac{\mathbf{T}_k^-(\bar{\theta}_{k,i})}{\bar{\phi}_k(\bar{\theta}_{k,i})}, \tag{5.26}$$

where we have collected the impurity parameters as follows

$$\{\theta_{k,i}\} \equiv \Big\{ x_{k,i}^{(+)} + \xi_k^{(+)} \tfrac{\mathrm{i}}{2} \Big\} \cup \Big\{ x_{k,i}^{(-)} + \xi_k^{(-)} \tfrac{\mathrm{i}}{2} \Big\} \cup \Big\{ \omega_{k,i}^{\mathrm{z}} + \mathrm{i}\epsilon - \tfrac{\mathrm{i}}{2} \Big\} \cup \Big\{ \bar{\omega}_{k,i}^{\mathrm{p}} - \mathrm{i}\epsilon + \tfrac{\mathrm{i}}{2} \Big\}, \tag{5.27}$$

along with their complex conjugates $\{\bar{\theta}_{k,i}\}$.

The regularised equilibrium ensemble is naturally cast as a two-dimensional classical vertex model wrapped around a cylinder of circumference $L$ and height $2N$, with

$$N = \sum_k N_k = \sum_k \Big( n_k^{(+)} + n_k^{(-)} + n_k^{\mathrm{z}} + n_k^{\mathrm{p}} \Big), \tag{5.28}$$

as illustrated in Figure 2. The 'Boltzmann weights' are formally identified with the matrix elements of Lax matrices, through the Lax representation of transfer matrices provided in Eq. (2.33). By virtue of the involution property of Eq. (2.34), all stacking configurations of

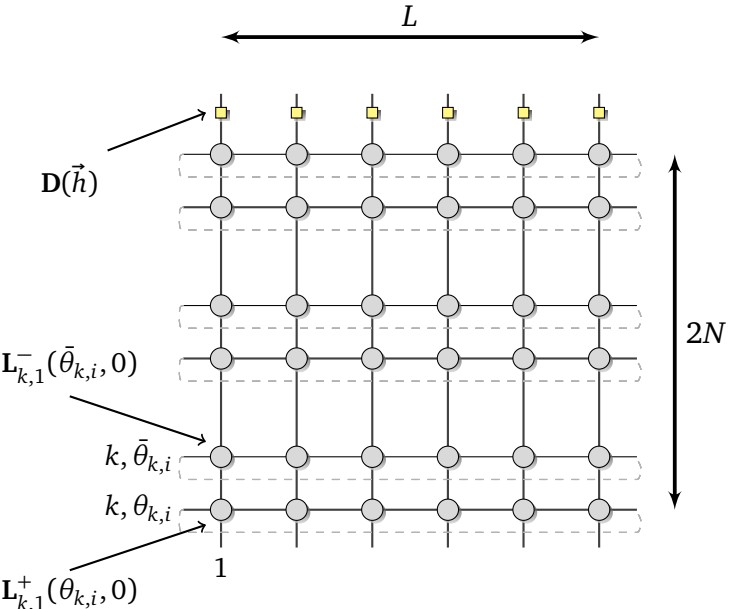

Figure 2: An illustration of the integrable two-dimensional vertex model corresponding to a regularised equilibrium macrostate, defined on a cylinder of circumference $L$ and height $2N$. The 'Boltzmann weights' are given by the matrix elements of the Lax matrices attached at each vertex. The square vertices attached at the vertical boundary incorporate the $\vec{h}$-dependent twist, and the dashed gray lines denote traces over the auxiliary spaces.

the row transfer matrices are equivalent. Let us emphasise that the resulting vertex model is not the common six-vertex model, but can be regarded instead as a fused variant thereof. Indeed, in general there is no upper bound on the 'number of vertices', as a generic equilibrium state requires arbitrary large spin representation labels. The contribution $\varrho_{\vec{h}}$ appears as an additional factor. It plays the role of a boundary twist when the cylinder is wrapped to a torus (i.e. when the ensemble is traced over).

## 6 The mirror system

The vertex-model regularisation of a generic equilibrium ensemble achieved in the previous section allows for a description of equilibrium states in manner which is complementary to the TBA analysis of Section 4. Here the free energy density is

$$f = -\lim_{L\to\infty}\lim_{N\to\infty}\frac{1}{L}\log\mathrm{Tr}_{\mathcal{H}}\left[\varrho_{\vec{h}}\prod_k\prod_{i=1}^{N_k}\frac{\mathbf{T}_k^+(\theta_{k,i})}{\phi_k(\theta_{k,i})}\frac{\mathbf{T}_k^-(\bar{\theta}_{k,i})}{\bar{\phi}_k(\bar{\theta}_{k,i})}\right],\tag{6.1}$$

which can be viewed as an inhomogeneous vertical iteration of row transfer matrices $\mathbf{T}_k$. Alternatively this can be re-expressed as a horizontal iteration of an inhomogeneous column transfer matrix, as illustrated in Figure 3. As the two-dimensional vertex model is homogeneous in the horizontal (i.e physical) direction, the iteration of the column transfer matrix is homogeneous, with the consequence that its dominant eigenvalue yields the free energy density. In this section we analyse this process in detail. We then pay particular attention to the re-emergence of the thermodynamic $\mathscr{Y}$-system, Eq. (4.11), in the large-$N$ limit.

Given a two-dimensional vertex model, we introduce the corresponding 'mirror system' as

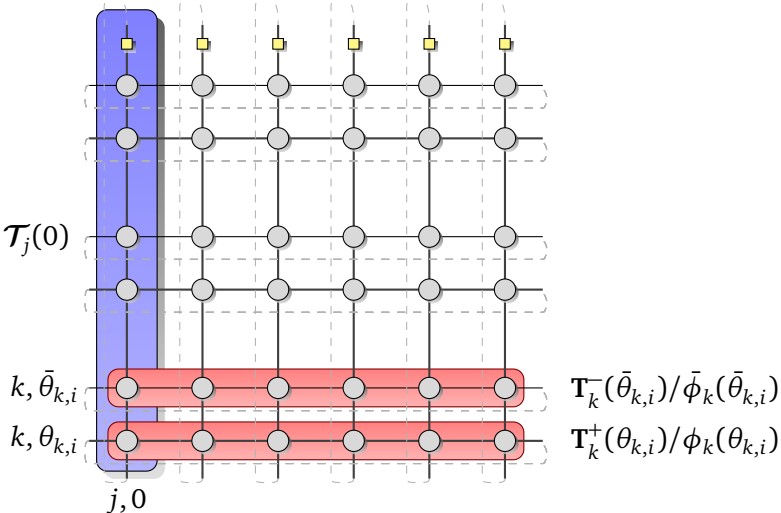

Figure 3: The equilibrium partition function is computed by wrapping the cylinder of Figure 2 to a torus, i.e. tracing over the physical sites. This can be viewed as an inhomogeneous iteration of the homogeneous row transfer matrices $\mathbf{T}_k^{\pm}$ (red) of the physical system as in Eq. (6.1). Alternatively it can be expressed as a homogeneous iteration of the inhomogeneous column transfer matrices $\mathcal{T}_j$ (blue) of the mirror system. The latter admits a dominant eigenvalue as given in Eq. (6.3).

the combination of the auxiliary spaces: $\mathcal{H}_{\mathrm{M}} \cong \bigotimes_k \mathcal{V}_k^{\otimes 2N_k}$. The fundamental column transfer matrix [2] acts on the mirror system as follows

$$\mathcal{T}_1(u) = \mathrm{Tr}_{\mathcal{V}_1}\left[ \mathbf{D}(\vec{h}) \bigotimes_k \bigotimes_{i=1}^{N_k} \frac{\mathbf{L}_{k,1}^+(\theta_{k,i}, u)}{\theta_{k,i} - u + (k+1)\frac{\mathrm{i}}{2}} \frac{\mathbf{L}_{k,1}^-(\bar{\theta}_{k,i}, u)}{\bar{\theta}_{k,i} - u - (k+1)\frac{\mathrm{i}}{2}} \right], \qquad (6.2)$$

with spectral parameter $u$, and the trace taken over the common fundamental space $\mathcal{V}_1$ of all the $\mathbf{L}_{k,1}$ inherited from a lattice site of the original spin chain. That is, in switching to the column transfer matrix the role of physical and auxiliary spaces is interchanged. We emphasise that unlike the row (physical) transfer matrices defined in Eq. (2.33), the column transfer matrix is inhomogeneous in terms of both impurities and representation labels, and has a boundary twist encoded by the factor $\mathbf{D}(\vec{h})$ which also acts on $\mathcal{V}_1$. For notational convenience we proceed by suppressing explicit dependence of $\mathcal{T}_1(u)$ on impurities $\theta_{k,i}$, twist $h$ and dimension $N$.

The fundamental column transfer matrix $\mathcal{T}_1(u)$ allows for computation of the free energy density through its dominant (i.e. largest) eigenvalue according to the prescription

$$f = -\lim_{N \to \infty} \lim_{L \to \infty} \frac{1}{L} \log \mathrm{Tr}_{\mathcal{H}_{\mathrm{M}}} \mathcal{T}_1(0)^L = -\lim_{N \to \infty} \log \widehat{\mathcal{T}}_1(0), \qquad (6.3)$$

where $\widehat{\mathcal{T}}_1(u)$ denotes the dominant eigenvalue. Care must be taken when interchanging the thermodynamic $L \to \infty$ and the scaling $N \to \infty$ limits (cf. [21, 22]), and in our analysis we justify this step by establishing full consistency, as summarised in Figure 1.

One way to proceed to determine $\widehat{\mathcal{T}}_1(0)$ is to obtain the Bethe equations which diagonalise $\mathcal{T}_1(u)$ as a route towards obtaining their spectrum $\mathcal{T}_1(u)$, and thereby its dominant eigenvalue. Here we instead take a different route and employ the framework of functional Bethe

---

[2]The canonical version of the column transfer matrix appears under several different names in the literature, including the 'virtual-space' [21], 'crossing' [23], 'column-to-column' [58], 'quantum' [27] and 'Matsubara' [59] transfer matrix.

ansatz [24,35,39,60], which conveniently allows for the mirror-system Bethe equations to be completely bypassed.

With aid of the fusion rule for the Lax matrices, we proceed by embedding $\mathcal{T}_1(u)$ into the infinite fusion hierarchy of commuting transfer matrices $\mathcal{T}_j(u)$. To this end, we extend the family of Lax matrices $\mathbf{L}_{k,1}(v,u)$ to $\mathbf{L}_{k,j}(v,u) : \mathcal{V}_k \otimes \mathcal{V}_j \to \mathcal{V}_k \otimes \mathcal{V}_j$. These are readily obtained through fusion [34,38,61,62]

$$\mathbf{L}_{k,j}(v,u) = \mathcal{N}_{k,j}^{-1}(v,u)\,\mathcal{P}_j \prod_{a=1}^{j} \mathbf{L}_{k,1}^{(a)}\big(v,u+(j+1-2a)\tfrac{\mathrm{i}}{2}\big)\mathcal{P}_j, \qquad (6.4)$$

where each of $j$ copies $\mathbf{L}_{k,1}^{(a)}$ acts on $\mathcal{V}_k \otimes \mathcal{V}_1^{(a)}$, $\mathcal{P}_j : \bigotimes_{a=1}^{j} \mathcal{V}_1^{(a)} \mapsto \mathcal{V}_j$ is the totally-symmetric projection operator,

$$\mathcal{P}_j = \prod_{a=1}^{j} \frac{\vec{\mathbf{S}} \cdot \vec{\mathbf{S}} - a(a-1)}{j(j+1) - a(a-1)}, \qquad (6.5)$$

with $\vec{\mathbf{S}} = \sum_{a=1}^{j} \vec{\mathbf{S}}^{(a)}$, and $\mathcal{N}_{k,j}(v,u) = \prod_{a=1}^{j-k}(v-u+(k-j-1+2a)\tfrac{\mathrm{i}}{2})$ is the common polynomial produced by fusion. Correspondingly, the higher-spin column transfer matrices are given explicitly as

$$\mathcal{T}_j(u) = \mathrm{Tr}_{\mathcal{V}_j} \left[ \mathbf{D}(\vec{h}) \bigotimes_k \bigotimes_{i=1}^{N_k} \frac{\mathbf{L}_{k,j}^+(\theta_{k,i},u)\mathbf{L}_{k,j}^-(\bar{\theta}_{k,i},u)}{\prod_{a=1}^{\min(j,k)}\big(\theta_{k,i}-u+(k-j+2a)\tfrac{\mathrm{i}}{2}\big)\big(\bar{\theta}_{k,i}-u-(k-j+2a)\tfrac{\mathrm{i}}{2}\big)} \right], \qquad (6.6)$$

where here $\mathbf{D}(\vec{h})$ acts on $\mathcal{V}_j$.

In particular, analogously to the physical $T$-functions Eq. (2.35), the hierarchy of column $\mathcal{T}$-functions $\mathcal{T}_j(u)$ satisfy the Hirota equation

$$\mathcal{T}_j^+ \mathcal{T}_j^- = \varphi_j \bar{\varphi}_j + \mathcal{T}_{j-1} \mathcal{T}_{j+1}, \qquad j \geq 0, \qquad (6.7)$$

with initial conditions $\mathcal{T}_{-1} \equiv 0$, $\mathcal{T}_0 = 1$, and the scalar potentials

$$\varphi_j = \prod_k \prod_{i=1}^{N_k} \Psi_{j,k}^{-1}(u - \theta_{k,i}), \qquad \bar{\varphi}_j = \prod_k \prod_{i=1}^{N_k} \Psi_{j,k}(u - \bar{\theta}_{k,i}), \qquad (6.8)$$

with $\Psi_{j,k}$ given by Eq. (2.29). By construction, $\mathcal{T}_j(u)$ are meromorphic functions of rapidity variable $u$. The boundary twist manifests itself as non-trivial large-$u$ asymptotics

$$\lim_{|u| \to \infty} \mathcal{T}_j(u) = \frac{\sinh((j+1)h/2)}{\sinh(h/2)} \equiv \chi_j(h), \qquad (6.9)$$

parametrised by $h = |\vec{h}|$ through the $\mathfrak{su}(2)$ characters $\chi_j(h)$ obeying the 'classical limit' of the Hirota equation

$$\chi_j^2(h) = 1 + \chi_{j-1}(h)\chi_{j+1}(h). \qquad (6.10)$$

The associated 'gauge'-invariant $\mathcal{Y}$-functions

$$\mathcal{Y}_j = \frac{\mathcal{T}_{j-1}\mathcal{T}_{j+1}}{\varphi_j \bar{\varphi}_j} = \frac{\mathcal{T}_j^+ \mathcal{T}_j^-}{\varphi_j \bar{\varphi}_j} - 1, \qquad (6.11)$$

are again meromorphic functions, obeying the canonical $\mathcal{Y}$-system relations [42]

$$\mathcal{Y}_j^+ \mathcal{Y}_j^- = \big(1 + \mathcal{Y}_{j-1}\big)\big(1 + \mathcal{Y}_{j+1}\big), \qquad (6.12)$$

with large-$u$ asymptotics

$$\mathcal{Y}_j^\infty \equiv \lim_{|u|\to\infty} \mathcal{Y}_j(u) = \chi_j^2(h) - 1. \tag{6.13}$$

The above functional relations, Eqs. (6.7), (6.11) and (6.12), are valid for the full spectrum of $\mathcal{T}_j(u)$ at finite $N$. To specify a particular eigenvalue, it is necessary to identify their analytic data in the physical strip $\mathcal{P}$. This is made transparent by transforming the functional relations to integral equations on the real rapidity axis. To perform this step, it is useful to introduce adapted $\mathcal{T}$-functions,

$$\mathcal{T}_j^{\text{adp}}(u) = \frac{\mathcal{T}_j(u)}{\prod_i \tau(u; t_{j,i}^z)}, \tag{6.14}$$

obtained by factoring out their (possible) zeros $t_{j,i}^z$ in $\mathcal{P}$, so that $\log \mathcal{T}_j^{\text{adp}}(u)$ are analytic on $\mathcal{P}$ with constant large-$u$ asymptotics. The absence of poles of $\mathcal{T}_j(u)$ in $\mathcal{P}$ can be seen from the denominator in Eq. (6.6). Now manipulating Eq. (6.11), by taking the logarithm and convolving with $s$, we obtain

$$\log \mathcal{T}_j = \sum_i \log \tau\left(u; t_{j,i}^z\right) + s \star \log\left(\varphi_j \bar{\varphi}_j\right) + s \star \log\left(1 + \mathcal{Y}_j\right). \tag{6.15}$$

We similarly transform the $\mathcal{Y}$-system relations, Eq. (6.12), to the real axis. The procedure is analogous to that described in Section 4, specialised to meromorphic data only. Introducing adapted $\mathcal{Y}$-functions

$$\mathcal{Y}_j^{\text{adp}}(u) = \mathcal{Y}_j(u) \frac{\prod_b \tau(u; y_{j,b}^p)}{\prod_a \tau(u; y_{j,a}^z)}, \tag{6.16}$$

by again multiplying out all (possible) zeros $y_{j,a}^z$ and poles $y_{j,b}^p$ on the physical the strip $\mathcal{P}$, so that $\log \mathcal{Y}_j^{\text{adp}}(u)$ are analytic on $\mathcal{P}$ with constant large-$u$ asymptotics. Substituting this into Eq. (6.12), taking the logarithm and convolving with $s$, we end up with the coupled integral equations

$$\log \mathcal{Y}_j = d_j + I_{j,\ell} s \star \log(1 + \mathcal{Y}_\ell), \tag{6.17}$$

with source terms

$$d_j = \sum_a \log \tau(u; y_{j,a}^z) - \sum_b \log \tau(u; y_{j,b}^p). \tag{6.18}$$

The spectrum of the $\mathcal{T}$-functions is thus given by Eq. (6.15), where the $\mathcal{Y}$-functions obey Eq. (6.17), and eigenvalues are specified by the locations of the analytic data $t_{j,a}^z$, $y_{j,a}^z$, $y_{j,a}^p$. To determine this state-dependent data, one has in general to resort to the Bethe equations. The $\mathcal{Y}$-functions however inherit certain zeros and poles directly from the impurity data through the scalar potentials in Eq. (6.11). To identify these we need the following properties which can verified from the definition of $\Psi_{j,k}$ in Eq. (2.29): if $w$ belongs to the lower-half of the strip $\mathcal{P}$ then $\Psi_{j,k}(u-w)$ has a zero at $u = w + \frac{i}{2}$ if and only if $j = k$, and has no poles in $\mathcal{P}$, while if $w$ belongs to the upper-half of the strip $\mathcal{P}$ then $\Psi_{j,k}(u-w)$ has a pole at $u = w - \frac{i}{2}$ if and only if $j = k$, and has no zeros in $\mathcal{P}$. Then from Eq. (6.8) one deduces that zeros of the $\mathcal{Y}$-functions are located at

$$\left\{ x_{k,i}^{(-)} + \left(1 + \xi_k^{(-)}\right)\tfrac{i}{2} \right\} \cup \left\{ \omega_{k,i}^z + i\epsilon \right\}, \tag{6.19}$$

along with their complex conjugates, and we remind that $\xi_k^{(-)} < 0$. Correspondingly, the poles of the $\mathcal{Y}$-functions appear at

$$\left\{ x_{k,i}^{(+)} + \left(1 - \xi_k^{(+)}\right)\tfrac{i}{2} \right\} \cup \left\{ \omega_{k,i}^p + i\epsilon \right\}, \tag{6.20}$$

along with their complex conjugates, with $\xi_k^{(+)} > 0$.

The eigenstate for which these are the only zeros and poles of the $\mathcal{Y}$-functions inside $\mathcal{P}$ is a *distinguished state* of the mirror system. In particular, the corresponding $\mathcal{T}$-functions $\mathcal{T}_j(u)$ are *analytic and free of zeros* in the strip $\mathcal{P}$ for this state, a further consequence of Eq. (6.11). As any zero of a $\mathcal{T}$-function in $\mathcal{P}$ yields a negative definite contribution to Eq. (6.15), i.e.

$$\log \tau(u; w) < 0, \qquad u \in \mathbb{R}, \ \ w \in \mathcal{P}, \tag{6.21}$$

it is natural to anticipate that this state gives the dominant eigenvalue determining the free energy density through Eq. (6.3). In the following we establish this assertion by taking the large-$N$ limit, and demonstrating that it reproduces precisely the TBA description of Section 4.

## 6.1 The scaling limit

In the preceding analysis we considered the mirror system at finite $N$. At this level, there is no strict distinction between the impurities encoding the node and analytic data from Eq. (5.27). Now we take the large-$N$ scaling limit in which this distinction becomes manifest. We focus on the distinguished state identified above for which all the $\mathcal{T}$-functions are analytic and free of zeros in the physical strip $\mathcal{P}$.

The task is to inspect the large-$N$ scaling limit for both Eq. (6.15) and Eq. (6.17). For Eq. (6.15) the non-trivial $N$-dependent term is $\log\left(\varphi_j \bar{\varphi}_j\right)$. For clarity, we consider below the contributions coming from the node and analytic data separately, that is we split

$$\log\left(\varphi_j \bar{\varphi}_j\right) = \left[\log\left(\varphi_j \bar{\varphi}_j\right)\right]_\lambda + \left[\log\left(\varphi_j \bar{\varphi}_j\right)\right]_\nu. \tag{6.22}$$

The node data $\lambda_j$ involves a sum over pairs of impurities

$$\left\{ x_{k,i}^{(\pm)} + \xi_k^{(\pm)} \tfrac{\mathrm{i}}{2}, \ x_{k,i}^{(\pm)} - \xi_k^{(\pm)} \tfrac{\mathrm{i}}{2} \right\}, \tag{6.23}$$

yielding

$$\left[\log\left(\varphi_j \bar{\varphi}_j\right)\right]_\lambda = \sum_{\sigma=+,-} \sum_k \sum_{i=1}^{n_k^{(\sigma)}} \log \left[ \frac{\Psi_{j,k}\left(u - x_{k,i}^{(\sigma)} + \xi_k^{(\sigma)} \tfrac{\mathrm{i}}{2}\right)}{\Psi_{j,k}\left(u - x_{k,i}^{(\sigma)} - \xi_k^{(\sigma)} \tfrac{\mathrm{i}}{2}\right)} \right]. \tag{6.24}$$

Individual contributions, which can be deduced with aid of Eq. (2.29), are of the form

$$-\frac{\Lambda_k^{(\pm)}}{n_k^{(\pm)}} G_{j,k}\left(u - x_{k,i}^{(\pm)}\right). \tag{6.25}$$

In the large-$N$ limit, the net contribution of Eq. (6.24) yields

$$\lim_{N \to \infty} \left[\log\left(\varphi_j \bar{\varphi}_j\right)\right]_\lambda = -G_{j,k} \star \lambda_k, \tag{6.26}$$

upon removing the large-rapidity cut-off $\Lambda^\infty$ and converting sums to convolution-type integrals. In the process, the impurities re-condense in accordance with Eq. (5.7).

The contribution from the analytic data stemming from zeros $\{\omega_{k,i}^{\mathrm{z}} + \mathrm{i}\epsilon - \tfrac{\mathrm{i}}{2}, \bar{\omega}_{k,i}^{\mathrm{z}} - \mathrm{i}\epsilon + \tfrac{\mathrm{i}}{2}\}$ and poles $\{\omega_{k,i}^{\mathrm{p}} + \mathrm{i}\epsilon + \tfrac{\mathrm{i}}{2}, \bar{\omega}_{k,i}^{\mathrm{p}} - \mathrm{i}\epsilon - \tfrac{\mathrm{i}}{2}\}$ inside $\mathcal{P}$, reads

$$\left[\log\left(\varphi_j \bar{\varphi}_j\right)\right]_\nu = -\sum_k \sum_{a=1}^{n_k^{\mathrm{z}}} \log \left[ \frac{\Psi_{j,k}^+\left(u - \omega_{k,a}^{\mathrm{z}} - \mathrm{i}\epsilon\right)}{\Psi_{j,k}^-\left(u - \bar{\omega}_{k,a}^{\mathrm{z}} + \mathrm{i}\epsilon\right)} \right] - \sum_k \sum_{b=1}^{n_k^{\mathrm{p}}} \log \left[ \frac{\Psi_{j,k}^-\left(u - \omega_{k,b}^{\mathrm{p}} - \mathrm{i}\epsilon\right)}{\Psi_{j,k}^+\left(u - \bar{\omega}_{k,b}^{\mathrm{p}} + \mathrm{i}\epsilon\right)} \right]. \tag{6.27}$$

Employing the identity $\log \Psi_{j,k}^{\pm} = \pm(1+K)_{j,k} \star \log \tau$ which follows from Eqs. (2.23) and (2.26), this becomes

$$[\log (\varphi_j \bar\varphi_j)]_\nu = -(1+K)_{j,\ell} \star \zeta_\ell^{\text{reg}}, \tag{6.28}$$

with $\zeta_j^{\text{reg}}$ given by Eq. (5.23). Here the infinitesimal regulator $\epsilon$ can be safely dropped. As the degrees of the Padé approximations tend to infinity with $N$, we then obtain

$$\lim_{N\to\infty} [\log (\varphi_j \bar\varphi_j)]_\nu = -(1+K)_{j,\ell} \star \zeta_\ell. \tag{6.29}$$

Thus combining the contributions from the node and analytic data we recover the TBA chemical potentials (4.16),

$$\lim_{N\to\infty} \log (\varphi_j \bar\varphi_j) = -\mu_j. \tag{6.30}$$

Next we take the large-$N$ scaling limit of Eq. (6.17). Here we must examine the source terms $d_j$ given by Eq. (6.18). The node data consists of pairs of zeros and poles, cf. Eq. (6.19),

$$\left\{ x_{k,i}^{(\pm)} + \left(1 \mp \xi_k^{(\pm)}\right) \tfrac{\text{i}}{2}, x_{k,i}^{(\pm)} - \left(1 \mp \xi_k^{(\pm)}\right) \tfrac{\text{i}}{2} \right\}, \tag{6.31}$$

each contributing to $d_j$ a term

$$\log\left[ \tau\left(u; x_{k,i}^{(\pm)} + \left(1 \mp \xi_k^{(\pm)}\right) \tfrac{\text{i}}{2}\right) \tau\left(u; x_{k,i}^{(\pm)} - \left(1 \mp \xi_k^{(\pm)}\right) \tfrac{\text{i}}{2}\right) \right]. \tag{6.32}$$

Since there are $n_k$ terms, in the $n_k \to \infty$ limit we are left with

$$\frac{\Lambda_k^{(\pm)}}{n_k^{(\pm)}} s\left(u - x_{k,i}^{(\pm)}\right), \tag{6.33}$$

employing Eq. (2.23). The full contribution to $d_j$ in the large-$N$ scaling limit then retrieves the node data, that is $s \star \lambda_j$. The contribution from the regularised analytic data, which is for finite $N$ straightforwardly given by $\zeta_j^{\text{reg}}$, and in the large-$N$ limit converges to $\zeta_j$. In conclusion,

$$\lim_{N\to\infty} d_j = d_j = s \star \lambda_j + \zeta_j, \tag{6.34}$$

are precisely the source terms (4.15) of the local form of the TBA equations Eq. (4.7).

**Free energy.** Having taken the continuum scaling limit, we can now determine the eigenvalues of $\mathcal{T}$-functions $\mathcal{T}_j(u)$ encoding the distinguished state. In particular, employing Eq. (6.30) in Eq. (6.15) we obtain

$$\log \mathcal{T}_1(0) = -s \circ \mu_1 + s \circ \log (1 + \mathcal{Y}_1). \tag{6.35}$$

Finally, substituting this into Eq. (6.3) as the dominant eigenvalue we re-obtain precisely the expression for the free energy given by Eq. (4.6). As the large-$N$ scaling limit of the canonical $\mathcal{Y}$-functions $\mathcal{Y}_j$ satisfy the local TBA equations Eq. (4.7) due to Eq. (6.34), we have fully recovered the TBA description of Section 4.

In conclusion,

> *for a general equilibrium state of the Heisenberg spin-1/2 chain, the thermodynamic $\mathscr{Y}$-functions coincide with the large-N scaling limit of the distinguished canonical $\mathcal{Y}$-functions associated with the dominant eigenvalue of the mirror system.*

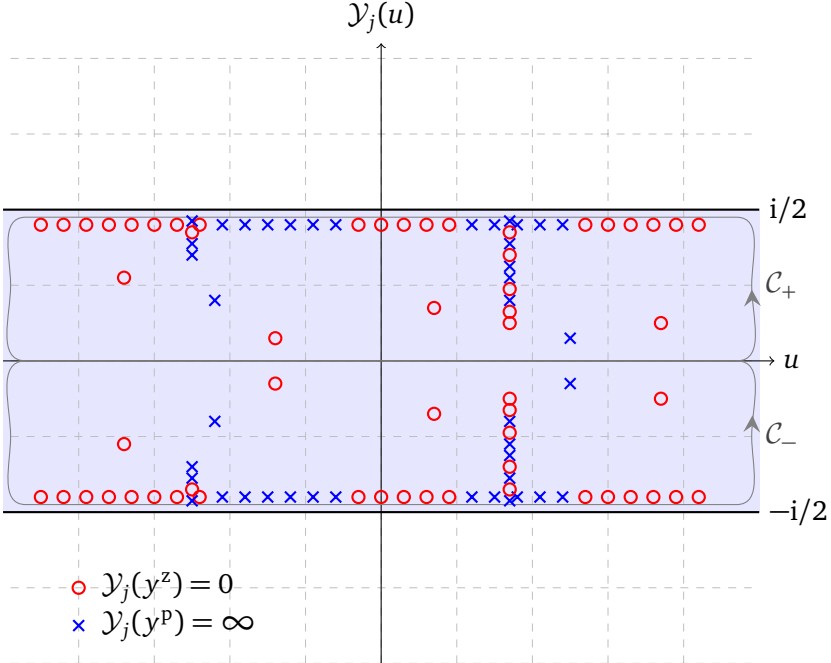

Figure 4: Schematic depiction of the analytic data for a typical meromorphic $\mathcal{Y}$-function associated with the dominant eigenvalue of the fundamental column transfer matrix $\mathcal{T}_1(u)$ at finite $N$. The analytic data comprises zeros (red circles) and poles (blue crosses) in the complex strip $\mathcal{P}$. The integration contours $\mathcal{C}_+$ and $\mathcal{C}_-$, encircling the upper and lower halves of the physical strip $\mathcal{P}$, are separated from the lines $u = \pm\frac{i}{2}$ by the regulator $\epsilon$. The contributions from the regularised node data $\lambda_j$ appear separated by distance $\xi_j^{(\pm)} = \left|\Lambda_j^{(\pm)}\right|/2\pi n_j^{(\pm)}$ from the lines $u = \pm\frac{i}{2}$. The analytic data lie within the contours $\mathcal{C}_\pm$, with clusters of zeros and poles appearing as Padé approximants of non- integer branch points.

## 7 Discussion

In the preceding sections we considered generic equilibrium states and developed the dual approaches of TBA and lattice regularisation, as summarised in Figure 1. The central role in establishing their equivalence is played by the mirror system, which provides a compact regularisation of a given equilibrium state. The mirror system exhibits a 'universal' canonical structure, seen here in Eqs. (6.7)-(6.13), which is deep-rooted in the fusion properties of the underlying quantum symmetry algebra. One of the key findings of this work is however that in thermodynamic/large-$N$ limit the canonical structure is superseded by an emergent equilibrium landscape, encoded in non-trivial node terms $\lambda_j$ and generically non- meromorphic thermodynamic $\mathscr{Y}$-functions involving branch points inside $\mathcal{P}$. There are several discernible aspects which deserve a brief discussion.

**Breakdown of the canonical $\mathcal{Y}$-system.**  The thermodynamic $\mathscr{Y}$-system generically contains state-dependent node terms $\lambda_j$, see Eq. (4.11). This non-canonical form, which is seemingly conflict with the expected universality of the canonical $\mathcal{Y}$-system [42], has been previously observed the context of quantum quenches where several explicit examples have been identified [7,16,63]. Here we clarify this aspect.

   The $\mathcal{Y}$-functions at finite $N$ obey the canonical $\mathcal{Y}$-system, Eq. (6.12), with $n_j^{(\pm)}$ zeros and

poles stemming from the regularised $\lambda_j$ residing are finite distance $\xi_j^{(\pm)} = |\Lambda_j^{(\pm)}|/2\pi n_j^{(\pm)}$ from the boundary of $\mathcal{P}$. A subtlety however arises in the thermodynamic scaling limit $N \to \infty$, where $\xi_j^{\pm}$ invariably become smaller than the positive infinitesimal regulator $\epsilon$ which controls the width of the physical strip $\mathcal{P}$ as in Eq. (2.22). [3] In this event, a subset of the analytic data leaves the nullspace of $s^{-1}$ by escaping from the integration contours $\mathcal{C}_\pm$ and finally collapsing onto the boundary of $\mathcal{P}$ at $\mathrm{Im}(u) = \frac{1}{2}$. This piece of information emerges through the recondensed $\lambda_j$ as in Eq. (6.34), rendering the thermodynamic $\mathscr{Y}$-system of the non-canonical form. Indeed, a simple example of a non-canonical $\mathscr{Y}$-system is provided by the canonical Gibbs equilibrium state, specified by $\lambda_j(u) = \pi J \beta \delta_{j,1}\delta(u)$ and $\zeta_j(u) = 0$. The corresponding TBA source term thus reads $d_j = s \star \lambda_j = \pi J \beta \delta_{j,1}s(u)$, in agreement with [26].

**Non-meromorphic $\mathscr{Y}$-functions.**  As observed in the TBA analysis, in general the thermodynamic $\mathscr{Y}$-functions are non-meromorphic complex functions. This is to be contrasted with the canonical $\mathcal{Y}$-functions of the mirror system describing regularised macrostates which are manifestly meromorphic on $\mathcal{P}$. The non-meromorphic structure appears in the large-$N$ scaling limit, when the analytic data from the interior of $\mathcal{P}$ form macroscopic condensates. A degree-$N$ Padé approximation of a generic non-meromorphic $\mathscr{Y}$-function in terms of a canonical (i.e. meromoprhic) $\mathcal{Y}$-function provides a particular algorithm of information compression.

**Non-canonical asymptotics.**  A third subtlety concerns the large-$u$ asymptotics of the thermodynamic $\mathscr{Y}$-functions. At finite $N$ the canonical $\mathcal{Y}$-functions possess a one-parameter family of large-$u$ asymptotics parametrised by $h$, cf. Eq. (6.13), expressed through the solution to the classical limit of the Hirota equation Eq. (6.10). In the large-$N$ limit however, upon removing the regulator $\Lambda^\infty \to \infty$, these asymptotics may decouple, with the $\mathscr{Y}$-functions acquiring non-canonical large-$u$ asymptotics provided the large-$u$ asymptotics of the chemical potentials $\mu_j(u)$ are non-trivial. A particular instance of this is an infinite-parametric family of 'dispersionless' states, that is the class of states characterised by constant $\mu_j$.

## 8 Conclusion

In this work we have undertaken a comprehensive study of the equilibrium landscape of the isotropic Heisenberg spin-1/2 chain. We have developed a robust and unified framework which encompasses both the Thermodynamic Bethe Ansatz and the two-dimensional vertex-model regularisation approaches to thermodynamics. In particular we have explained the emergence of a splitting of the chemical potentials $\mu_j$ into two contributions: the node data $\lambda_j$ which determine the thermodynamic $\mathscr{Y}$-system, and the analytic data $\zeta_j$ which encode the branch points of the $\mathscr{Y}$-functions in the physical strip $\mathcal{P}$. These characterise equilibrium states in distinct ways, endowing the equilibrium landscape with a structure that is deserving of further exploration.

There are several novel features of the work worth highlighting. Firstly, we express the generic density matrix Eq. (3.1) in a form which reflects the underlying $\mathfrak{su}(2)$ symmetry of the model, which clarifies how the polarisation of the mode operators $\boldsymbol{\rho}_j$ is set. In our TBA analysis, we demonstrate on general grounds that the equilibrium $\mathscr{Y}$-functions are generically non-meromorphic in the physical strip $\mathcal{P}$. Our lattice regularisation of a generic ensemble treats the node and analytic data separately. For the node data we invoke a discretisation of the $\lambda_j(u)$, and achieve a variant of 'Trotterisation' without actually appealing to the Suzuki-Trotter formula. For the analytic data we develop a contour integration procedure, and intro-

---

[3]We remind that in distinction to $N$, $\epsilon$ is *not* an adjustable parameter of the regularisation scheme.

duce Padé approximants to regularise generic branch points of $\mathscr{Y}$-functions. In Section 6 we reconnect the TBA and vertex model approaches by identifying a distinguished eigenvalue of the mirror system, and use it to demonstrate complete equivalence of the descriptions. The interconnected nature of our analysis is summarised in Figure 1.

A technical point emerging from our study is an explanation of the breakdown of the canonical $\mathcal{Y}$-system, i.e. the emergence of the thermodynamic $\mathscr{Y}$-system. Put simply, this is due to competition between the infinitesimal regulator $\epsilon \equiv 0^+$ which is tied to locality of the $\rho_j$, and the regularisation parameter $N$ which is finite for any vertex model approximation of the ensemble and diverges in the scaling limit. At finite $N$ all the poles and zeros of the $\mathcal{Y}_j$ lie on the physical strip $\mathcal{P}$ of Eq. (2.22) and the corresponding $\mathcal{Y}$-system is canonical, Eq. (6.12). The poles and zeros resulting from the node data $\lambda_j$ however approach the boundary of the strip as $1/N$, so that in the large-$N$ limit they escape $\mathcal{P}$, resulting in the appearance of node terms in the thermodynamic $\mathscr{Y}$-system, Eq. (4.11).

Throughout the work we have adopted a universal language, which should facilitate extensions to other models solvable through the Bethe ansatz framework. The simplest generalisation of the model considered here is its uniaxial anisotropic deformation [19]. The extension to the 'hyperbolic' (easy-axis) regime, with quantum deformation parameter $q \in \mathbb{R}$, is straightforward and readily follows from an analytic continuation of the local degrees of freedom (i.e. the scattering matrix) of the isotropic model considered here. In distinction, in the 'trigonometric' (easy-plane) regime with the deformation parameter $q = e^{i\gamma}$ at the roots-of-unity $\gamma = m/\ell \in \mathbb{R}$, the total number of local degrees of freedom depends quite intricately on the values of co-prime integers $m$ and $\ell$, see [20, 40, 64]. To implement our approach, the compete set of unitary local charges is required, already identified previously in [7]. The closure of the functional hierarchy and the string-charge duality requires here an extra family of non-unitary charges [44, 65, 66].

Going forward, an important open question is whether there exist physically discernible features associated with the structures identified here. Addressing this will require analysing the correlation functions of local observables [47, 59, 67–69]. We anticipate that it will be interesting to examine the splitting between node and analytic data in this context, and hope that the framework we provide offers a solid foundation upon which to proceed.

**Acknowledgements.** We thank O. Gamayun for valuable remarks on the manuscript. E. I. acknowledges support by VENI grant number 680-47-454 by the Netherlands Organisation for Scientific Research (NWO). E. Q. acknowledges support from ANR IDTODQG project grant ANR-16-CE91-0009 of the French Agence Nationale de la Recherche, the Foundation for Fundamental Research on Matter (FOM), the Netherlands Organization for Scientific Research (NWO), and the European Research Council under ERC Advanced grant 743032 DYNAMINT.

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
