# Peer review of "The equilibrium landscape of the Heisenberg spin chain"

_SciPost Physics, doi:SciPost Phys. 7, 033 (2019)_

## Round 2 · Referee Report · Anonymous (Referee 1) · 2019-5-25

Strengths

  1. Good presentation
  2. Valid results
  3. A number of new technical details
  4. Important explanation of the breakdown of the $Y$-system

Weaknesses

  1. The usage of expressions for some concepts and methods does not always follow the traditions, and thus can be difficult to understand, or even misleading. (will be explained below)
  2. The discussion of earlier literature is not sufficient. The relevant works are cited, but the connection to them is not worked out properly. (examples will be given below)
  3. A technical point: the discussion of the $\varepsilon$ regulator is not precise and not clear enough. (concrete problems with this will be explained below)

Report

This paper deals with a characterization of the equilibrium states of the XXZ spin chain. It is an extension of earlier work of the authors, and here the new idea is to work out in full detail the Quantum Transfer Matrix (QTM) or mirror channel description of the states and ensembles, and to give a direct link to the usual Thermodynamic Bethe Ansatz description. In doing this a number of important technical details are worked out, which are new. (for example the so-called Trotter limit of the QTM is treated here in a way which is different from the usual methodology, here the authors use a discretization in some integrals for the ensembles, and not a formal Suzuki-Trotter decomposition) Also, the explanation of the breaking of the $Y$-system, and its mechanism, is important, and partially new.

I think that the results are valid, and they are important additions to the general picture regarding equilibrium ensembles of the spin chain.

Nevertheless I have a number of comments.

-The title of the paper includes the expression ''the equilibrium landscape''. With all due respect, to me this seems like a sensationalist use of words. Generally I don't like the idea to use new words for existing concepts, especially when the new additions to the concept itself do not change the main meaning. Here I don't see any new addition to the main concepts: the authors are describing equilibrium ensembles of the XXZ model. So why use a new word for this? The TBA description using Bethe root densities has existed since the early days.
-In the introduction it is stated that ''To our knowledge, no previous work on the statistical mechanics of exactly solvable mod- els, including a large body of work on solvable classical vertex models, has achieved a similar classification of the entire equilibrium manifold of a model.''. This is not clear to me. Even earlier works of the authors [4,7,45] have dealt with this exact problem, not to mention the standard TBA description. As far as I understand, the authors are adding new characterization in the ''mirror channel'', but this does not imply that the earlier works were not ''similar classifications''. With all due respect, this is again somewhat sensationalist.
-The authors use a lattice discretization of the ensembles. They call this the ''mirror system''. Now this is a name which I believe comes from the AdS/CFT literature, to be used for something that has existed since the early days, and early works of Bazhanov, Lukyanov, Zamolodchikov, Klümper, Pearce, and possibly many others. I think that the ''mirror system'' name can be used, but other traditional names should be included too. For example in the works of the Wuppertal group (Boos, Göhmann, Klümper, etc) this is consistently called the ''quantum channel'' and the associated transfer matrix the ''Quantum Transfer Matrix''. These names are not mentioned, but they should be, at least once. Also, in Sec. 6 there are no citations to any of these works.
-The idea to connect the QTM to the TBA appeared in some works earlier, and regarding the XXZ chain the first explicit discussion was in [26]. The paper [26] does very similar manipulations: they also use a functional Bethe Ansatz to connect the two descriptions. This is also used in the present paper. The idea is mentioned on page 17, but here a reference should be made to [26] which did the same thing for thermal ensembles.
-The regulator $\varepsilon$ is mentioned at multiple places. It plays a very important role. For every finite $\varepsilon$ the $\rho$ operators are quasi-local, but their range increases as $\varepsilon$ decreases. If I understand it correctly, in the strict $\varepsilon\to 0$ limit the quasi-locality property is lost. I think that this is not clearly stated, although it is implicitly mentioned. A more clear statement should be made, for example at the end of Section 2. Similarly, at the bottom of page 22 the explanation is not so clear. On the one hand side it is mentioned that $\varepsilon$ is infinitesimal, which means that it should be smaller than anything else, and we should be talking about a $\varepsilon\to 0$ limit, on the other hand it is also mentioned that the $\xi_j^\pm$ parameters eventually become smaller than $\varepsilon$. This is meaningless as a mathematical statement, even though we can imagine what it is supposed to mean. So I think that a more clear and mathematically precise explanation should be given at this point too. If I understand it correctly, this has to do with the exchange of the $\varepsilon\to 0$ and $N\to \infty$ limits. This should be made more clear.
-Regarding the breaking of the $Y$-system, the explanation is good, but the mathematical phenomenon should be explained more clearly. At every finite $N$ the $Y$-system has to hold. But we observe that it is broken in the $N\to\infty$ limit. In parallel with going to the details of the mechanism, it should be explained: Does the sequence of $Y$ functions converge pointwise? Or where does it converge? And what is the precise mathematical statement which leads to the breaking of the $Y$-system? Is it true that the operations of ''pointwise limit of the sequence of $Y$ functions'' and ''analytical continuation to the boundary of the physical strip'' do not commute? Or something similar? This should be made very clear. The statements about the movement and condensation of poles are important, but this is a detailed technical explanation, technical description, and the mathematical essence is not clear enough yet. And the summary of this explanation could be added to the Conclusions too. After all, many experts understand that the $Y$-system should hold generally, and that in the TBA for generic states we do not have it. So the breaking of the $Y$-system is in fact an important phenomenon.
-In Section 3 the Eigenstate Thermalization Hypothesis (or a generalized version) is mentioned, but its validity for the XXZ chain is not explained. The missing point is that the local equivalence of states with the same charges, so same root densities is not explained. The papers [59-63] dealing with correlation functions are cited in the Conclusions, but they should be mentioned also in Section 3, with a clear explanation, that in the thermodynamic limit the local observables only depend on the root densities.
-In (4.15) we have a finite sum, or at least it seems to be a finite sum. How do we know that this is the case? Or can it be infinite? Can we have an integral over these functions, with some constant $\alpha$? It would then correspond to some branch cuts. Is this possible, or excluded?

Some further small comments are also given below.

I think the paper can be published after a minor revision, once my comments are addressed.

Requested changes

-I suggest to avoid the expression ''the equilibrium landscape'' in the title. In my opinion ''equilibrium ensembles'' would be more precise, clear, and not sensationalist.
-Tone down the introduction regarding the novelty of this work (paragraph 3).
-Add the conventional expressions for the lattice discretization, for example ''Quantum Transfer Matrix method'', at least once. Also, cite the correct references at these points, in Sec. 6.
-Explain that the paper [26] did very similar manipulations in the thermal case.
-Clarify the meaning and implications of the $\varepsilon$ regulator. Explain more clearly the locality of $\rho$, and the exchange of the $\varepsilon\to 0$ and $N\to \infty$ limits.
-Elaborate on the breaking of the $Y$-system, and the mathematical phenomenon. (see questions and comments above)
-Elaborate on the meaning of the ETH in Section 3, and add here the references dealing with local correlation functions.
-The introduction states that ''and raises the question whether interactions induce physically discernible features among equilibrium states. The objective of this paper is to establish a framework to address this''. This is not clear to me, please explain this better. What are physically discernible features? If we are talking about observables, local correlators, or spectral functions then this paper does not deal with them, neither with entanglement. If the authors are thinking about a future possibility to deal with correlators, then they should cite here the existing works dealing with correlations, for example [59-63].
-I think that a comma and a ''be'' are missing here: ''the partition function can [be] regularised through a two-dimensional classical vertex model[,] where''
-At the beginning of Section 2 it should be mentioned that with a positive J this is a ferromagnetic spin chain.
-Eq. (2.15) deals with the Yang-Yang entropy. Here the Yang-Yang paper should be cited, or perhaps also some review article or book dealing with this, maybe Takahashi's book on thermodynamics of spin chains.
-Clarify the summation in (4.15). What are the possibilities and why?
-At the end of Section 5 please mention that the vertex model is essentially the fused 6-vertex model. It is already mentioned that this is not the usual 6-vertex model, and that the spin variables can be arbitrarily big. This is of course correct. But it should be mentioned (or repeated) nevertheless that this comes from the fused model.

  • validity: top
  • significance: high
  • originality: good
  • clarity: high
  • formatting: perfect
  • grammar: perfect

Author:  Enej Ilievski  on 2019-07-22  [id 570]

(in reply to Report 1 on 2019-05-25)
Category:
answer to question

We are grateful to the Referee for their assessment of our manuscript and for their remarks. Let us address each point in turn.

The referee writes:

The title of the paper includes the expression ''the equilibrium landscape''. With all due respect, to me this seems like a sensationalist use of words. Generally I don't like the idea to use new words for existing concepts, especially when the new additions to the concept itself do not change the main meaning. Here I don't see any new addition to the main concepts: the authors are describing equilibrium ensembles of the XXZ model. So why use a new word for this? The TBA description using Bethe root densities has existed since the early days.

Our response: In this work we achieve an explicit vertex model representation of Generalized Gibbs ensembles, including the complete classification of the entire manifold of macrostates and reconciliation of two complementary computational approaches. This is to be contrasted with the previous studies which are devoted to particular families of states relevant in various nonequilibrium physical scenarios. Here our focus the attention exclusively to the formal structure of the manifold. The objective was thus to find a title which would efficiently grasp the underlying philosophy, avoiding unnecessary technical jargon if possible. The word "landscape" which came to mind serves this purpose quite well and, despite the suggestion by the referee, we remain convinced that this choice is adequate. We do not see it as sensationalist, but instead view it as an appropriate succinct term to reflect the essence of the manuscript. Moreover it needs to be emphasised that we go well beyond the 'TBA description using Bethe root densities (which) has existed since the early days'.

The referee writes:

In the introduction it is stated that ''To our knowledge, no previous work on the statistical mechanics of exactly solvable models, including a large body of work on solvable classical vertex models, has achieved a similar classification of the entire equilibrium manifold of a model.''. This is not clear to me. Even earlier works of the authors [4,7,45] have dealt with this exact problem, not to mention the standard TBA description. As far as I understand, the authors are adding new characterization in the ''mirror channel'', but this does not imply that the earlier works were not ''similar classifications''. With all due respect, this is again somewhat sensationalist.

Our response: We have modified the Introduction to clarify what we meant here, and we hope the Referee finds it satisfactory. Our objective is to place the work in context and to highlight clearly what we have achieved.

The referee writes:

The authors use a lattice discretization of the ensembles. They call this the ''mirror system''. Now this is a name which I believe comes from the AdS/CFT literature, to be used for something that has existed since the early days, and early works of Bazhanov, Lukyanov, Zamolodchikov, Kl\"{u}mper, Pearce, and possibly many others. I think that the ''mirror system'' name can be used, but other traditional names should be included too. For example in the works of the Wuppertal group (Boos, G\"{o}hmann, Kl\"{u}mper, etc) this is consistently called the ''quantum channel'' and the associated transfer matrix the ''Quantum Transfer Matrix''. These names are not mentioned, but they should be, at least once. Also, in Sec. 6 there are no citations to any of these works.

Our response: There is indeed ambiguity regarding the language, which is perhaps to be expected given the long and rich history of the subject. We have added a footnote in Sec. 6 to reflect this. Vertex model constructions using commuting transfer matrices date back to the pioneering works by Baxter and others, which includes "row", "column", "diagonal" and "corner" transfer matrices among others. These concepts gradually dispersed into other domains of statistical physics and other branches of physics which accordingly adapted the language. For instance, in the context of quantum spin chain one find papers "virtual" and "crossing" transfer matrices prior the above mentioned "quantum'' transfer matrix. In a more recent work by renowned mathematical physicists even the name "Matsubara" channel has been used. In our perspective, we deal here with the lattice manifestation of the widely known ``mirror transformation" which can be traced back to works of Zamolodchikov, which permits one to interpret imaginary times as the spatial direction of a fictitious interacting spin model. We find this picture physically most suggestive and this motivates our usage.

The referee writes:

The idea to connect the QTM to the TBA appeared in some works earlier, and regarding the XXZ chain the first explicit discussion was in [26]. The paper [26] does very similar manipulations: they also use a functional Bethe Ansatz to connect the two descriptions. This is also used in the present paper. The idea is mentioned on page 17, but here a reference should be made to [26] which did the same thing for thermal ensembles.

Our response: We thank the Referee for this remark. We have added a paragraph to the Introduction to highlight this connection.

The referee writes:

The regulator $\epsilon$ is mentioned at multiple places. It plays a very important role. For every finite $\epsilon$ the $\boldsymbol{\varrho}$ operators are quasi-local, but their range increases as $\epsilon$ decreases. If I understand it correctly, in the strict $\epsilon \to 0$ limit the quasi-locality property is lost. I think that this is not clearly stated, although it is implicitly mentioned. A more clear statement should be made, for example at the end of Section 2. Similarly, at the bottom of page 22 the explanation is not so clear. On the one hand side it is mentioned that $\epsilon$ is infinitesimal, which means that it should be smaller than anything else, and we should be talking about a $\epsilon \to 0$ limit, on the other hand it is also mentioned that the $\xi_{\pm j}$ parameters eventually become smaller than $\epsilon$. This is meaningless as a mathematical statement, even though we can imagine what it is supposed to mean. So I think that a more clear and mathematically precise explanation should be given at this point too. If I understand it correctly, this has to do with the exchange of the $\epsilon \to 0$ and $N \to \infty$ limits. This should be made more clear.

Our response: Let us be clear and stress that we do not ever operate with the $\epsilon \to 0$ limit. We have added a footnote in Section 7 to emphasise this. We have also added more comments around the introduction of $\epsilon\equiv0^+$ in Section 2 which we hope clarify its use in the manuscript. As $\epsilon$ is strictly positive the quasi-locality of the $\boldsymbol{\rho}$ is not lost, and we have re-emphasised this in the text.

The referee writes:

Regarding the breaking of the Y-system, the explanation is good, but the mathematical phenomenon should be explained more clearly. At every finite N the Y-system has to hold. But we observe that it is broken in the $N\to \infty$ limit. In parallel with going to the details of the mechanism, it should be explained: Does the sequence of Y functions converge pointwise? Or where does it converge? And what is the precise mathematical statement which leads to the breaking of the Y-system? Is it true that the operations of ''pointwise limit of the sequence of Y functions'' and ''analytical continuation to the boundary of the physical strip'' do not commute? Or something similar? This should be made very clear. The statements about the movement and condensation of poles are important, but this is a detailed technical explanation, technical description, and the mathematical essence is not clear enough yet. And the summary of this explanation could be added to the Conclusions too. After all, many experts understand that the Y-system should hold generally, and that in the TBA for generic states we do not have it. So the breaking of the Y-system is in fact an important phenomenon.

Our response: We thank the Referee for this suggestion and have included such a summary in the Conclusion, which we hope clarifies the `competition' between $\epsilon$ and $1/N$.

We have not attempted a mathematical analysis of convergence properties. This was our priority and we expect this will mostly be of interest to mathematically minded experts. Convergence in the pointwise sense may be expected, but we would be unwise to speculate on this before undertaking a careful analysis. Establishing convergence properties of Pad\'{e} approximants, which have been proposed as a tool for regularizing non-meromorphic $Y$-functions, often turn out pretty nasty.

The referee writes:

In Section 3 the Eigenstate Thermalization Hypothesis (or a generalized version) is mentioned, but its validity for the XXZ chain is not explained. The missing point is that the local equivalence of states with the same charges, so same root densities is not explained. The papers [59-63] dealing with correlation functions are cited in the Conclusions, but they should be mentioned also in Section 3, with a clear explanation, that in the thermodynamic limit the local observables only depend on the root densities.

Our response: We agree and have added this.

The referee writes:

In (4.15) we have a finite sum, or at least it seems to be a finite sum. How do we know that this is the case? Or can it be infinite? Can we have an integral over these functions, with some constant $\alpha$? It would then correspond to some branch cuts. Is this possible, or excluded?

Our response: We have added some remarks after Eq. (4.15) to clarify this aspect. The sum has to involve a finite number of terms in order to guarantee a well-defined source term. Infinite convergent sequences are however also allowed, an example being convergent sequences of Pad\'{e} approximated $Y$-functions where simple poles and zeros condense into branch cuts. We adopt the convention that all branch cuts extend vertically away from the real axis.

The referee writes:

The introduction states that ``and raises the question whether interactions induce physically discernible features among equilibrium states. The objective of this paper is to establish a framework to address this''. This is not clear to me, please explain this better. What are physically discernible features? If we are talking about observables, local correlators, or spectral functions then this paper does not deal with them, neither with entanglement. If the authors are thinking about a future possibility to deal with correlators, then they should cite here the existing works dealing with correlations, for example [59-63].

Our response: The objective of this work was to work out the formalism, and the quoted sentenced merely serves as a motivation. We do not have any applications in mind yet. Physically discernible features will be the subject of the future work.

---

## Round 2 · Referee Report · Anonymous (Referee 2) · 2019-6-11

Strengths

  1. Explicit construction of the QTM-like object for an arbitrary equilibrium ensemble
  2. Comparison with the TBA approach

Weaknesses

  1. Lack of the context in the presentation, in particular concerning the QTM approach for the thermal equilibrium.

Report

The main result of the paper The equilibrium landscape of the Heisenberg spin chain'' is the systematic construction of the QTM-like object to describe any given equilibrium ensemble of the XXX spin chains (the authors call it themirror system''). The author also show the equivalence between this mirror system and the TBA approach. The result is new and very important, it is well known how the QTM approach simplify the study of the spin chains at finite temperature, it can be expected that similar simplifications can be achieved through the mirror system presented in this paper. For this reason I am sure that this paper should be accepted.

Unfortunately in the current version the authors skip almost completely the existing context and in particular the quantum transfer matrix (QTM) approach to the thermal equilibrium and corresponding non-linear integral equations (NLIE). It is very strange that the QTM is never mentioned in the paper while the approach is very similar. Even if several important papers on the QTM are cited by the authors (very briefly and only in the introduction) they never really discuss the relation between their approach (based on discretisation) and standard QTM (based on Suzuki-Trotter formula). The equivalence with the TBA approach established by the authors can be also compared with a similar procedure introduced in [26]. It would be interesting also to show that for the simplest case of the thermal equilibrium the mirror system introduced by the authors is equivalent to the usual QTM [25].

Some technical details should be also clarified, the most important one is the infinitesimal parameter $\epsilon$ as it is used throughout the paper and appears in the definition of the quantum d'Alembertian (which is unusual). It is introduced first in the definition of the left-inverse of the $s$-kernel and as far as I understand the r.h.s. of the equation (2.20) can be rather written as $f(u+\frac i2-i0)+f(u-\frac i2 +i0)$ to avoid poles (by the way there should be some analyticity conditions for $f$). It is not clear then how the physical strip $\mathcal{P}$ is defined. I think that the parameter $\epsilon$ needs some clarification.

Second subtle point is the introduction of the cut-off $\Lambda^{\infty}$. The natural question arising here if the possibility to introduce such a cut-off for the lattice regularisation imposes some condition on behaviour of chemical potentials $\mu_j(u)$?

There are also some minor notes:

  • Introducing string solutions (2.8) the authors claim that corrections are exponentially small. In fact it is true for a finite number of strings and not true when the strings are distributed with some density.

  • The star convolution is introduced in a tensor product of function spaces (2.16) but then it is used in a single space (2.20), (the meaning is clear but notations should be clarified).

  • The term "quantum Hirota equation" (2.35) seems to be an abuse of language, it is just the Hirota equation.

  • Introducing the algebraic formulation of integrability the founding paper should be cited:

L.D. Faddeev, E.K. Sklyanin and L.A. Takhtajan, Quantum inverse problem method I, Theor. Math. Phys. 40, 688 (1979),

  • The citation to the paper where fusion procedure was first introduced should be added P.P. Kulish, N. Reshetikhin and E. Sklyanin, Yang-Baxter equation and representation theory I Lett. Math. Phys. 5, 393 (1981),

  • In the context of NLIE (or functional Bethe ansatz) the Destri De Vega paper should be cited

C. Destri and H.J. DeVega, New thermodynamic Bethe Ansatz equations without strings, Phys. Rev. Lett 69 2313 (1992),

I would also suggest to add appropriate citations in the text of the technical introduction (section 2). Most of the relevant papers are in the reference list but should be correctly cited in the text where the corresponding results and methods are introduced.

Requested changes

see report

  • validity: high
  • significance: high
  • originality: high
  • clarity: high
  • formatting: excellent
  • grammar: excellent

Author:  Enej Ilievski  on 2019-07-22  [id 569]

(in reply to Report 2 on 2019-06-11)
Category:
answer to question

We are grateful to the referee for their report and for their suggestions and remarks.

The referee writes:

Unfortunately in the current version the authors skip almost completely the existing context and in particular the quantum transfer matrix (QTM) approach to the thermal equilibrium and corresponding non-linear integral equations (NLIE). It is very strange that the QTM is never mentioned in the paper while the approach is very similar. Even if several important papers on the QTM are cited by the authors (very briefly and only in the introduction) they never really discuss the relation between their approach (based on discretisation) and standard QTM (based on Suzuki-Trotter formula). The equivalence with the TBA approach established by the authors can be also compared with a similar procedure introduced in [26]. It would be interesting also to show that for the simplest case of the thermal equilibrium the mirror system introduced by the authors is equivalent to the usual QTM [25].

Our response: This was an oversight on our part and we now make explicit reference to QTM in the revised version. We have also added a paragraph to the Introduction to highlight the connection to Ref [26], and have also added a comment in Sec. 6 to highlight equivalence with the QTM treatment of the Gibbs state.

The referee writes:

Some technical details should be also clarified, the most important one is the infinitesimal parameter $\epsilon$ as it is used throughout the paper and appears in the definition of the quantum d'Alembertian (which is unusual). It is introduced first in the definition of the left-inverse of the s-kernel and as far as I understand the r.h.s. of the equation (2.20) can be rather written as $f(u+i/2-i 0) + f(u-i/2 + i 0)$ to avoid poles (by the way there should be some analyticity conditions for $f$). It is not clear then how the physical strip $\mathcal{P}$ is defined. I think that the parameter $\epsilon$ needs some clarification.

Our response: This detail is primarily one of notation, and we hope it is now clarified in the revised version. Due to its importance we attach a symbol to the positive infinitesimal $\epsilon\equiv 0^+$. Our primary reason for doing this is to emphasise the common origin of all instances of $0^+$ as it plays a crucial role throughout our analysis, from ensuring locality of the charges to being instrumental in the breakdown of the canonical Y-system. For this reason also we highlight its explicit appearance into the definition of the physical strip. Moreover we feel that the explicit introduction of the the symbol $\epsilon$ makes the technical treatment in Section 6 more transparent as not doing so and instead using $0^+$ throughout creates a level of ambiguity. In the revised manuscript we have clarified the introduction of $\epsilon$ in Section 2, and have commented upon its inclusion in the definition of $\mathcal{P}$.

The referee writes:

Second subtle point is the introduction of the cut-off $\Lambda_{\infty}$. The natural question arising here if the possibility to introduce such a cut-off for the lattice regularisation imposes some condition on behaviour of chemical potentials $\mu_{j}(u)$?

Our response: The rapidity cut-off $\Lambda_{\infty}$ is necessary for the outlined regularization. Chemical potential $\mu_{j}(u)$ may in generally possess a constant asymptotic value at large-$u$, in which case the macrostate can be systematically approximated upon removing the cut-off $\Lambda_{\infty}$ and taking that large-$N$ scaling limit. This is commented upon in Section 7.

The referee writes:

Introducing string solutions (2.8) the authors claim that corrections are exponentially small. In fact it is true for a finite number of strings and not true when the strings are distributed with some density.

Our response: We agree that this is a subtle point. We have amended our statement in the text.

---

## Round 3 · Author Response

Dear editor,

We have revised the manuscript by following the suggestions by the referees.
We hope that the manuscript is ready for publication in SciPost.

The authors.

---

## Round 3 · List of Changes

- To better clarify the context of our work, we have rephrased and extended the introduction.
We mention in particular the approach based on the Quantum Transfer Matrix non-linear integral equations (inclunding a reference to Destri and de Vega), and connections to previous work connecting QTM and TBA.
- We highlight $J>0$ corresponds to the ferromagnetic spin chain.
- We softened our statement concerning the `string hypothesis'.
- We added the definition for the scalar convolution integral in Eq. (2.14).
- We clarified the introduction of the infinitesimal regulator $\epsilon\equiv0^+$, and why we include it in our definition of the physical strip.
- In the end of Section 2, we improved the explanations regarding the meaning of the regulator $\epsilon$ for both the local
charges $\mathbf{X}_{j}(v)$ and the mode density operators $\boldsymbol{\rho}_{j}(u)$.
- In the introduction to Section 3, we now mention that local correlation functions are functional of the quasi-particle densities and cite a few relevant papers.
- We have added clarifying remarks after Eq. (4.15).
- In Section 5 we comment that the general vertex model can be regarded as a fused 6-vertex model.
- In Section 6 we have added a footnote on naming conventions for the column transfer matrix.
- In Section 7, we added a footnote regarding the regulator $\epsilon$ and write out explicitly the TBA source terms of the canonical Gibbs state, matching that of ref. [26].
- A summary of the explanation for the breakdown of the canonical $Y$-system is added to the Conclusion in Section 8.

---

## Editorial Decision

published